# NOT JUST PRETTY PICTURES: TOWARD INTERVENTIONAL DATA AUGMENTATION USING TEXT-TO-IMAGE GENERATORS

## ABSTRACT

Neural image classifiers are known to undergo severe performance degradation when exposed to inputs that exhibit covariate shifts with respect to the training distribution. A general interventional data augmentation (IDA) mechanism that simulates arbitrary interventions over spurious variables has often been conjectured as a theoretical solution to this problem and approximated to varying degrees of success. In this work, we study how well modern Text-to-Image (T2I) generators and associated image editing techniques can solve the problem of IDA. We experiment across a diverse collection of benchmarks in domain generalization, ablating across key dimensions of T2I generation, including interventional prompts, conditioning mechanisms, and post-hoc filtering, showing that it substantially outperforms previously state-of-the-art image augmentation techniques independently of how each dimension is configured. We discuss the comparative advantages of using T2I for image editing versus synthesis, also finding that a simple retrieval baseline presents a surprisingly effective alternative, which raises interesting questions about how generative models should be evaluated in the context of domain generalization.

## 1 INTRODUCTION

The success of deep image classifiers is largely built on the assumption that the train and test data come from the same domain – i.e., that they are independent and identically distributed (i.i.d.) – but in real-world applications, small changes in the environmental conditions under which the image is captured can break this assumption, significantly degrading their performance (Gulrajani & Lopez-Paz, 2020; Wang et al., 2022a; Sakaridis et al., 2020). Since these changes only affect the inputs (i.e., the covariates) in some features without altering the labels, this form of distribution shift is also known as covariate-shift (Quionero-Candela et al., 2009). The employment of complex augmentation pipelines integrating image transformation primitives has been one of the most effective approaches for training classifiers that generalize to domains with environmental conditions that differ from their training data (Cugu et al., 2022; Hendrycks et al., 2022; 2020; Zhao et al., 2020; Qiao et al., 2020; Zhou et al., 2020).

Many types of augmentation primitives can be thought as reproducing (often approximately) a controlled and targeted manipulation of the domain-specific environmental conditions in which the image was captured (e.g., illumination or weather conditions) without affecting the label-related features. As such, augmentations may be understood as an automated, low-cost way of simulating interventions over the environmental factors that are likely to change across domains, turning *observational data* (i.e., with no intentional manipulation of the environment) into approximated *interventional data* (Ilse et al., 2021; Wang et al., 2022b). Motivated by this principle, several works have theoretically conjectured the utility of an augmentation mechanism capable of simulating arbitrary interventions (Ilse et al., 2021; Wang et al., 2022c; Wang & Veitch, 2022; Gowda et al., 2021). However, since it is not possible to target arbitrary interventions in the context of traditional augmentation pipelines (e.g., it is not possible to hard-code a pixel-space intervention to transform images of paintings into realistic photos), prior work has instead focused on leveraging prior knowledge about specific invariances expected to hold in the target domains (Hong et al., 2021; Li et al., 2020; Ilse et al., 2021) or targeting specific downstream applications (Ouyang et al., 2022; Gowda et al., 2021).

With the emergence of sophisticated open-source Text-to-Image (T2I) generators like Stable Diffusion (Rombach et al., 2021) that can be used to generate and edit images using text prompts describing the desired output image, a natural question is whether such generators could be used as a *general-purpose* interventional data augmentation (IDA) mechanism.

Unlike previous approaches, these models can be used off-the-shelf without requiring manual hard-coding of individual interventions or training on application-specific data: instead, it is only necessary to describe the desired intervention via language (e.g., prompting the model with image-editing prompts such as "a photo taken at night" or "it is a cloudy day" could simulate corresponding interventions over lighting conditions). Several recent works have studied the usefulness of synthetic data from T2I generators (see, e.g., Bansal & Grover, 2023; Azizi et al., 2023; He et al., 2023; Trabucco et al., 2023), but using them to simulate interventions for

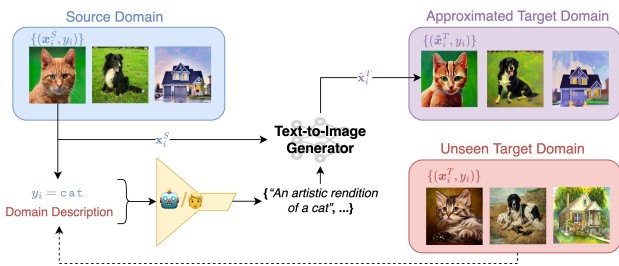

**Figure 1: Using Text-to-Image (T2I) Generators as general-purpose Interventional Data Augmentation mechanisms.** Given input images and editing instructions, T2I generators can manipulate them to approximate the desired transformation. Based on the prompt given, the generator edits the image to resemble the target domain. The resulting manipulated images can be used to train more robust and generalizable models.

data augmentation has not yet been studied. In this work we systematically analyze the extent to which current T2I generators can serve as general-purpose IDA mechanisms. We perform extensive experiments across several benchmarks for two key tasks in which the environmental and causal variables can be disentangled and the utility of synthetic interventional data can be precisely measured, Single Domain Generalization (SDG) and Reducing Reliance on Spurious Features (RRSF). We find that Stable Diffusion is highly effective for both tasks, substantially outperforming previously state-of-the-art data augmentation baselines across several widely used benchmarks of different complexity and scale, demonstrating that T2I generators are capable of implementing IDA across these important applications.

Our primary contributions are as follows:

- We carry out the first investigation of T2I generators as general-purpose interventional data augmentation mechanisms, showing their potential across diverse target domains and potential downstream applications.
- We perform extensive analyses over key dimensions of T2I generation, finding the conditioning mechanism to be the most important factor in IDA.
- We show that interventional prompts are also important to IDA performance; but in in contrast with previous works, we find that post-hoc filtering is not consistently beneficial.

## 2 PROBLEM SETTING AND RELATED WORKS

**The Problem of Out-of-Domain Generalization** Given a data distribution $P(\mathbf{x}, y) = P(y|\mathbf{x})P(\mathbf{x})$ where $\mathbf{x} \in \mathcal{X} \subset \mathbb{R}^d$, $y \in \mathcal{K} = \{1, 2, \ldots, |\mathcal{K}|\}$, learning a classifier amounts to estimating $\hat{f}(\mathbf{x}) \approx P(y|\mathbf{x})$ (i.e., predicting the conditional distribution of the label $y$ given a covariate $\mathbf{x}$) using a labelled training set $\mathcal{D}_{\text{train}} = \{(\mathbf{x}_i, y_i)\}_{i=1}^N$. Given the finite amount of data available in $\mathcal{D}_{\text{train}}$ and the high dimensionality of $\mathcal{X}$, the samples in $\mathcal{D}_{\text{train}}$ are not representative of the whole input space (i.e. $\mathbf{x}_i \in \mathcal{X}_{\text{train}} \subset \mathcal{X}$). When deployed in the wild, the classifier will likely be exposed to inputs sampled from regions of the input space not represented in the training set, even when $\mathcal{K}$ is the same. Specifically, we are in presence of *covariate-shift*, a form of distribution-shift. It has been empirically observed neural classifier's performance significantly degrades in the presence of covariate-shifted evaluation data (Gulrajani & Lopez-Paz, 2020; Wang et al., 2022a; Sakaridis et al., 2020).

Several theoretical frameworks have been developed to make the problem of out-of-domain (OOD) generalization well-posed (Wang et al., 2022b; Wang & Veitch, 2022; Ilse et al., 2021; Quionero-Candela et al., 2009 – in this work, we default to the framework proposed by Ilse et al. 2021). In computer vision, the core principle is that pixel values of image $\mathbf{x}_i \in \mathcal{X}$ are the result of a data generation process that combines (unobserved) features $h_{y_i}$ and $h_{\mathbf{c}_i}$ generated by the label $y_i$, and

conditions described by a vector of *environmental variables* $\mathbf{c}_i \in \mathcal{C}$ (Gowda et al., 2021; Ilse et al., 2021). To make the problem more tractable, it is often assumed it is possible to partition $\mathcal{C}$ into $M$ domains (i.e., $\mathcal{C} = \bigcup_{j=1}^{M} \mathcal{C}^j, \mathcal{C}^k \cap \mathcal{C}^h = \varnothing, \forall k \neq h$) based on how similarly the environmental conditions impact $\mathbf{x}$, so that the contextual variables values and impact are summarised in the discrete indices $j$ (Arjovsky et al., 2019). For instance, environmental variables could be aggregated to represent similar illumination conditions or backgrounds. Furthermore, an unobserved spurious confounder $\mathbf{s}$ might correlate both $y_i$ and $\mathbf{c}_i$. A high-performing classifier is likely to learn these spurious correlations, as they are predictive of the label $y_i$; but such correlations will (by definition) not hold under all environmental conditions, damaging classifiers' ability to generalize (Xiao et al., 2021; Geirhos et al., 2019).

**Simulating Interventional Data for Out-of-Domain Generalization** Existing works Ilse et al. (2021); Wang et al. (2022b) have proposed that such problem is solvable by performing interventions on $\mathbf{c}_i$ (i.e., manipulating $\mathbf{c}_i$ to break such spurious correlations without changing $y_i$). However, direct collection of interventional data is usually quite difficult (e.g., collecting datasets portraying the same object in all environments of interest may be highly impractical).

Identifying heuristic ways to disentangle causal from environmental factors has been a key component of several domain generalization methods. For example, CIRL (Lv et al., 2022) and ACVC (Cugu et al., 2022) manipulate the amplitude component of the image frequency spectrum of the Fourier transform, which is believed to approximately encode environmental information. Others, such as Hong et al. (2021); Li et al. (2020); Jackson et al. (2019), have tried using style transfer techniques in order to perturb the environmental factors while preserving the image content. Wang et al. (2022b) suggests using a Cycle-GAN that would preserve the causal factors in a cyclic transformation between domains with different styles. Beyond methods explicitly attempting to disentangling these two components, Ilse et al. (2021); Gowda et al. (2021) understand augmentations as simulating alterations of $\mathbf{c}_i$ without affecting $y_i$ – for example, rotations encode the belief that change of viewpoints should preserve the class label. Noticeably, these assumptions might not hold in all applications: for example, in digit classification, rotations of more than 90 degrees can swap the ground-truth labels of 6 and 9; so augmentations cannot be indiscriminately applied. Domain-agnostic data augmentation pipelines (such as those proposed by Hendrycks et al. 2020; Cubuk et al. 2020; DeVries & Taylor 2017; Hendrycks et al. 2022; Cugu et al. 2022) can be understood as hard-coding interventions on specific types of non-causal features that are expected to vary in novel environments. These assumptions may not hold across all possible domains. For this reason, Ilse et al. (2021) suggests a mechanism to select parametric hand-crafted augmentations that will likely affect mostly the environmental factors and not the causal factors.

**Text-to-Image Generators** With the recent rise of powerful, flexible T2I generative models (see, e.g., Nichol et al., 2021; Rombach et al., 2021; Ramesh et al., 2021), a natural question is whether these models, which are capable of zero-shot editing of arbitrary input images using natural-language prompts, could represent an effective IDA mechanism. Indeed, while some hand-crafted parametric augmentations can be straightforwardly implemented by a programmer to manipulate the image directly in pixel-space (e.g., lens distortion, chromatic aberration, vignetting, etc.), in many cases it is only possible to approximate them (often with much greater difficulty of implementation; e.g., introducing realistic rain or snow) or even impossible to reliably approximate (e.g., turning a cartoon into a photo, seamlessly changing the background of a scene, or modifying the material of an object). On the other hand, modern T2I models allow one to specify such transformations using natural language and have been observed to produce high quality samples (Meng et al., 2021). Their usage would be extremely convenient as these features are often provided off-the-shelf, after the generator providers have performed extensive training on large amounts of weakly supervised data, without requiring that one perform any task-specific fine-tuning (i.e., they can be used zero-shot).

**Data Augmentation with T2I Generators** Recent work has investigated how T2I models can be used to synthesize large-scale pre-training data (He et al., 2023; Sariyildiz et al., 2023; Azizi et al., 2023), compensate for the lack of training data in data-scarce environments (He et al., 2023; Trabucco et al., 2023), diagnose classifiers' lack of robustness to covariate-shift (Vendrow et al., 2023), and improve their robustness to some forms of distribution shift (Bansal & Grover, 2023). Closest to our work, Bansal & Grover (2023) show it is possible to use Stable Diffusion to generate synthetic data that improves the robustness of classifiers trained on ImageNet-1K (Deng et al., 2009) for some forms of covariate-shift (while reducing it on others) using the ensemble of prompts used to render ImageNet-1K zero-shot classification with CLIP more robust. A few of those prompts can be understood as implicitly performing IDA on the photo style. In this work, we depart from standard

ImageNet analyses in order to develop a deeper understanding of how T2I generators can be used as a general-purpose IDA mechanism by focusing on SDG and RRSF, allowing us to directly measure the effectiveness of T2I-simulated interventions in these settings across variable conditioning, prompting, and filtering techniques.

# 3 TEXT-TO-IMAGE GENERATORS FOR INTERVENTIONAL DATA AUGMENTATION

## 3.1 EXPERIMENTAL SETTING

Given some source training domain $\mathcal{D}^S = \mathcal{X}^S \times \mathcal{K}$ and some target domain $\mathcal{D}^T = \mathcal{X}^T \times \mathcal{K}$, our goal is to use T2I generators to approximately modify environmental features $\mathbf{c}_i$ to simulate $\mathcal{D}^T$ while keeping causal features for $y_i$ constant. For an image $\mathbf{x}_i^S \in \mathcal{X}^S$ of class $y_i$, we aim to transform it into $\hat{\mathbf{x}}_i^T$ such that it retains $y_i$ but looks like it has been sampled from $\mathcal{X}^T$, using SDEdit (Meng et al., 2021).

Among the editing techniques we consider, all of them are conditioned on a natural language prompt (we indicate as $\mathbf{z}_i^T$). For instance, if $\mathbf{x}_i^S$ represents a cartoon cat and $T = $ painting, $\mathbf{z}_i^T$ might be "a painting of a cat". Using the generator $G$, we transform $\mathbf{x}_i^S$ into $\hat{\mathbf{x}}_i^T = G(\mathbf{x}_i^S, \mathbf{z}_i^T)$. For all experiments, we use Stable Diffusion[1] (Rombach et al., 2021) pre-trained on LAION-Aesthetics.[2] Successful transformations produce $\hat{\mathbf{x}}_i^T \in \hat{\mathcal{X}}^T$ with $\hat{\mathcal{X}}^T \approx \mathcal{X}^T$. The synthetic pairs $\hat{\mathcal{D}}^T = (\hat{\mathbf{x}}_i^T, y_i)$ are then combined with $\mathcal{D}^S$ to perform ERM via cross-entropy minimisation to train the neural classifier (ResNet-18 and ResNet-50).

In this work, we focus on *Single-Domain Generalization* (SDG) and *Reducing Reliance on Spurious Features* (RRSF) as representative settings where access to a high-quality, general-purpose IDA mechanism is likely to have a substantial positive impact when training on both $\mathcal{D}^S$ and $\hat{\mathcal{D}}^T$. We describe our experimental formulation of both problems below.

**Single-Domain Generalization (SDG)** Given data $\mathcal{D}^S$ from a source domain accessible at training time, the goal of SDG is to achieve high performance on a set of datasets $\mathcal{D}^{T_j}$ with $j = 1, 2, ..., J$ sampled from different target domains (Qiao et al., 2020). In this setting, the generator uses $\mathcal{D}^S$ and $\mathbf{z}_i^{T_j}$ to generate $\hat{\mathcal{D}}^{T_j} \approx \mathcal{D}^{T_j}$. Following the standard evaluation procedure, we train a classifier on a single domain of each benchmark ($\mathcal{D}^S$) and test it on the others ($\mathcal{D}^{T_j}$), and report the average accuracy over the $J$ target domains. For our experiments, we consider four widely used benchmarks that vary for type of domain shift, number of classes and training samples: (1) **PACS** (Li et al., 2017), containing the domains art painting, cartoon, sketch, and photo; (2) **Office-Home** (Venkateswara et al., 2017), containing Art, Clipart, Product, and Real World; and (3) **NICO++** (Zhang et al., 2022), containing autumn, dim, outdoor, grass, water, and rock; and (4) **DomainNet** (Peng et al., 2019), containing clipart, infograph, painting, quickdraw, real, sketch. We provide a detailed description of the training procedure and other aspects of the SDG experiments in Appendix A.

**Reducing Reliance on Spurious Features (RRSF)** Sometimes the training data is collected in a domain $\mathcal{D}^S$ in which spurious features correlate with the labels. If a classifier relies on such spurious features, it will not be able to generalize to unseen test domains in which the spurious feature is no more predictive of the label (Xiao et al., 2021; Geirhos et al., 2019). In this setting, the prompts $\mathbf{z}_i^T$ intentionally perturb the spurious features to simulate domains in which the spurious correlation is broken. We consider three standard benchmarks: (1) **ImageNet-9** (Xiao et al., 2020) measures the over-reliance on background to predict the foreground (Background Bias), (2) **Cue-Conflict Stimuli (CCS)** (Geirhos et al., 2018) assesses the over-reliance on texture (Texture Bias), and (3) a subset of **CelebA**(Xiao et al., 2020) evaluates over-reliance on spurious demographic features (Demographic Bias – in this case, the spurious correlation between hair colour and gender in CelebA). See Appendix B for further details about each dataset and associated indices used to measure each form of bias.

---

[1]Specifically, we use Stable Diffusion 1.5, available here, following this implementation of SDEdit.

[2]Note that Stable Diffusion's pre-training dataset, LAION-Aesthetics, is a subset of the LAION-5B dataset (Schuhmann et al., 2022a) consisting of web-scraped text-image pairs with high "aesthetic scores" (Schuhmann et al., 2022b).

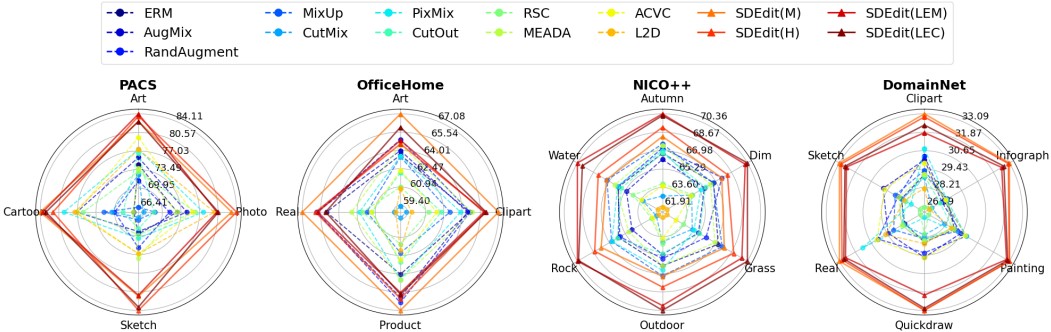

**Figure 2: Single Domain Generalization (SDG) Results.** Average SDG test accuracies on the remaining target domains when training ResNet-50 on each source domain (indicated on each axis) using the respective data augmentation methods. Baseline methods are visualized with dashed lines, and `SDEdit` methods with solid lines.

## 3.2 RESULTS

To evaluate the effectiveness of synthetic data from T2I models in IDA, we compare their results with important baselines broadly representing different approaches in the domain generalization literature (in addition to ERM): (1) **AugMix** (Hendrycks et al., 2020), (2) **RandAugment** (Cubuk et al., 2020), (3) **CutOut** (DeVries & Taylor, 2017), and (4) **PixMix** (Hendrycks et al., 2022), which all combine parametric transformations in complex pipelines to enhance model robustness. We also evaluate (5) **ACVC** (Cugu et al., 2022) combining parametric transformations and augmentations in the Fourier domain for style mixing. We also include interpolation-based methods like (6) **MixUp** (Zhang et al., 2017), (7) **CutMix** (Yun et al., 2019), methods learning generators that augment the training input to diversify it from the training domain (8) **L2D** (Wang et al., 2021b), and adversarial data augmentation techniques like (9) **MEADA** (Zhao et al., 2020) and (10) **RSC** (Huang et al., 2020).

**Single Domain Generalization** Considering the fact that Stable Diffusion is built on top of a text encoder that, like many LLMs, can be sensitive to small differences in prompts that are not generally meaningful to humans (see, e.g., Ribeiro et al., 2020; Wang et al., 2021a; Moradi & Samwald, 2021), we experiment with four distinct prompting strategies using `SDEdit` to measure its sensitivity to variation in prompts: (1) *Minimal (M)*, sentences including only the domain label, class label, and function words (i.e., articles or prepositions) as necessary to make the prompt grammatically correct, like "a `domain` of a `class`" (e.g., "a sketch of an elephant"); (2) *Domain expert (H)*, a collection of "hand-crafted" prompts authored by a human given only metadata descriptions provided by the respective

**Table 1: Average SDG Performance.** The number reported is the average Single Domain Generalization average of all domains in each dataset, each serving as a single source domain. The best and second best performing methods are highlighted with bold and underline, respectively.

|  | PACS | OfficeHome | NICO++ | DomainNet | Average |
|---|---|---|---|---|---|
| ERM | 61.96 | 61.94 | 69.95 | 25.26 | 54.78 |
| MixUp | 58.17 | 60.46 | 70.63 | 25.49 | 53.69 |
| CutMix | 58.50 | 57.16 | 67.03 | 24.47 | 51.79 |
| AugMix | 64.63 | 62.60 | 68.81 | 26.20 | 55.56 |
| RandAugment | 62.61 | 63.02 | 69.88 | 26.17 | 55.42 |
| CutOut | 60.87 | 60.03 | 69.23 | 24.90 | 53.76 |
| RSC | 64.58 | 59.10 | 67.37 | 23.32 | 53.59 |
| MEADA | 64.04 | 62.08 | 69.89 | 25.26 | 55.32 |
| PixMix | 67.12 | 61.43 | 69.48 | 25.53 | 55.89 |
| L2D | 68.89 | 58.37 | 65.19 | 24.75 | 54.30 |
| ACVC | 67.98 | 59.92 | 66.92 | 26.46 | 55.32 |
| SDEdit(M) | 76.43 | **64.66** | 71.12 | **31.94** | 61.04 |
| SDEdit(H) | **77.87** | 63.27 | 71.95 | 31.82 | 61.23 |
| SDEdit(LEC) | 76.38 | 63.43 | **73.69** | 31.44 | **61.24** |
| SDEdit(LEM) | 75.65 | 63.14 | 73.61 | 30.94 | 60.84 |

benchmarks, without looking at any samples form the target domain; and (3 & 4) *Language enhancement (LE)*, a collection of prompts generated by T5 (Raffel et al., 2020), in two variants: one that deterministically selects the highest-probability interventional prompts (LE$_C$), the other

that favors diversity in prompting $(\text{LE}_\text{M})$.[3] (See Appendix C for further details on each prompting strategy.)

As shown in Fig. 2, `SDEdit` outperforms all baselines regardless of the source domain (when averaging over target domains). Specifically, across all the considered benchmarks, using ResNet-50 with minimal prompt yields a $5\%$ improvement over the strongest baesline, PixMix, which in turn outperforms ERM by just $1.10\%$. We find that, when considering the performance on individual benchmarks, no single baseline consistently outperforms the others. This reinforces the observation that each of these techniques encodes different assumptions about the types of invariances expected to hold in the test domain. For the largest-scale dataset we consider, `DomainNet`, traditional data augmentation methods fail to demonstrate a substantial performance boost compared to ERM; but `SDEdit` is able to deliver a strong average performance boost of $5.48\%$. Comparing across all SDG datasets, `SDEdit` is the only method that consistently outperforms ERM across all benchmarks. (For a more detailed breakdown of all results figures, see Appendix A.)

We also find that is not usually the case that the most sophisticated prompting strategy performs best: in `PACS`, `OfficeHome` and `DomainNet`, the Minimal (M) and Handcrafted (H) strategies outperform $\text{LE}_\text{M}$ and $\text{LE}_\text{C}$, indicating that including additional details (e.g., specifying various styles of paintings across multiple prompts) does not yield obvious benefits, and may even be degrading performance, e.g., "injecting noise" into the pipeline. However, in `NICO++`, $\text{LE}_\text{M}$ and $\text{LE}_\text{C}$ show superior performance to (M) and (H), which may be explained by the fact the domain labels for NICO are not detailed enough for minimal prompts to be fully descriptive, meaning that the additional details included in prompts can be more beneficial in such contexts (as we observe here).

**Describing the Target Domain Is Not Necessary.** As noted above, Stable Diffusion is trained on a massive pre-training corpus of weakly-supervised data scraped from the web, which means it has likely been trained on samples that resemble a number of the considered test distributions. By comparison, while the baselines we consider do make limited assumptions about the type of interventions they perform (and therefore yield better or worse performance depending on whether those interventions correspond to the covariate shift from the source domain to test domain – see our RRSF analysis below), they do not have comparable access to approximations of the test domain. For this reason, we perform an experiment to "level the playing field" in order to better assess the usefulness of `SDEdit` as an in-

**Table 2:** SDG PACS result with ResNet-50. Columns are individual source domains; accuracies are the average test accuracy of the three remaining target domains when training using the indicated source domain. The lower part of the table highlights the comparison between accessing ($\checkmark$) or not accessing ($\times$) synthetic target domains.

|  | Art | Photo | Sketch | Cartoon | Average |
|---|---|---|---|---|---|
| ERM | 74.44 | 48.78 | 50.89 | 73.74 | 61.96 |
| MixUp | 66.31 | 42.98 | 45.64 | 77.76 | 58.17 |
| CutMix | 72.53 | 40.03 | 44.72 | 76.72 | 58.50 |
| AugMix | 75.80 | 51.32 | 49.99 | 81.42 | 64.63 |
| RandAugment | 71.38 | 46.80 | 55.95 | 76.33 | 62.61 |
| CutOut | 76.67 | 42.69 | 48.93 | 75.2 | 60.87 |
| RSC | 73.15 | 53.47 | 51.11 | 80.58 | 64.58 |
| MEADA | 73.72 | 48.78 | 59.81 | 73.84 | 64.04 |
| PixMix | 77.33 | 55.58 | 52.42 | 83.15 | 67.12 |
| L2D | 77.33 | 58.41 | 58.14 | 81.70 | 68.89 |
| ACVC | 79.63 | 52.76 | 58.13 | 81.40 | 67.98 |
| SDEdit(M) $\times$ | 81.21 | 57.54 | 80.60 | 84.76 | 76.03 |
| SDEdit(M) $\checkmark$ | 82.67 | 62.94 | 73.78 | 86.33 | 76.43 |

terventional mechanism by avoiding generating data resembling the test domain. Given a single training domain from the original dataset and a chosen test domain, we use `SDEdit` to transform the training data to all domains *except the test domain* ( `SDEdit`(M)$\times$), use it for IDA training and measure the accuracy on the test domain. Fixed a test domain, we repeat this experiment for each possible choice of the training domain, and report the average accuracy on the held-out test domain. In this case, we are measuring `SDEdit`'s capacity to perform IDA for SDG even when knowledge about the chosen test domain is not used in synthesizing interventional data. We find that the generative model-based methods still substantially outperform the data augmentation baselines, with only a marginal drop in performance with respect to the case in which the target domain is approximated by Stable Diffusion (`SDEdit`(M)$\checkmark$). This indicates the intervention on the environmental variables

---

[3]Note that, by design, none of the prompting strategies are optimized to boost the reported metrics: they are generated in a way that is independent from classifiers' performance on downstream tasks or the structure of the generator. See Appendix J for a complete list of image-generation prompts used in experiments.

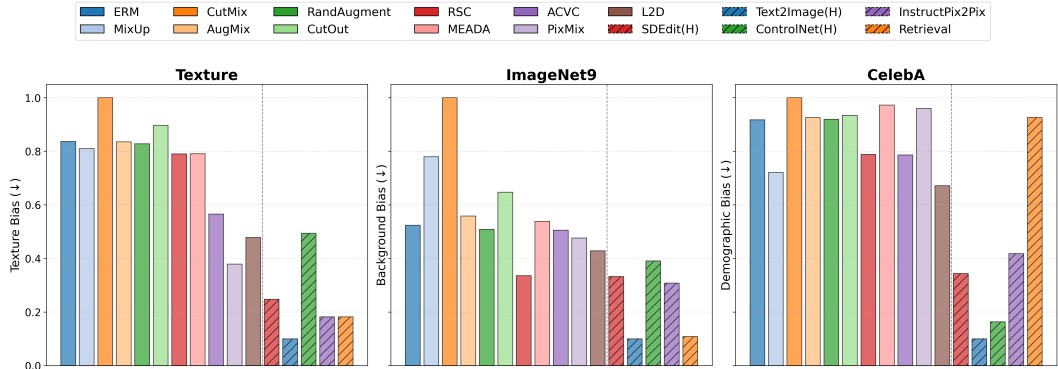

**Figure 3: Performance on Breaking Spurious Correlations.** Reliance on different image attributes in comparison with baselines (solid lines) and OURS (dash lines) using ResNet-18. (Lower scores are better.)

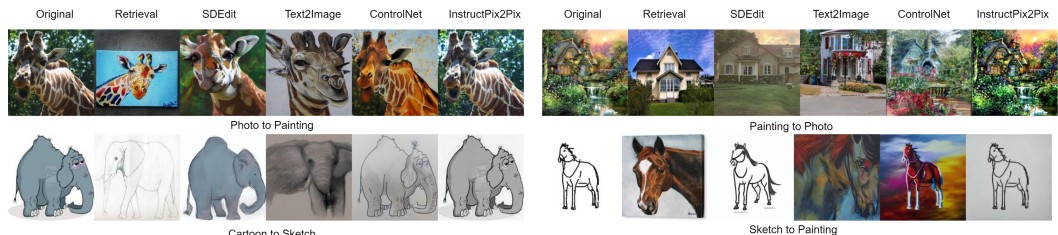

**Figure 4: Visualization of selected samples from PACS.** Recall that `Retrieval` and `Text2Image` do not take the `Original` image into account, but `SDEdit`, `ControlNet`, and `InstructPix2Pix` do.

produced by T2I generators aids in identifying invariant features that hold across domains that were not simulated by the generators themselves.

**Reducing Reliance on Spurious Features** Depending on the type of spurious correlation to be addressed in each experiment, we prompt `SDEdit` in different ways: for ImageNet-9 experiments, we handcraft prompts that describe a wide variety of possible backgrounds and randomise the combination of the object classes and backgrounds; for CCS, we use prompts that induce the generator to change the texture of the objects (e.g., turning them into a sculpture of a specific material); and for Celeb-A, we randomise the correlation between gender and hair colour. Our results are displayed in Fig. 3. We find that, although several techniques are often assumed to perturb spurious features in a way that is agnostic to the target domain, our experiments indicate that this may not the case – instead, baselines are (perhaps unsurprisingly) most effective when their augmentation pipeline implicitly intervenes over the corresponding spurious dependency. For example, PixMix mixes the input images with fractals that alter their texture (and often the background), but yields a worse Demographic Bias than ERM. In contrast, `SDEdit` can perform the desired augmentation based on the relevant spurious dependency by simply describing it using interventional prompts, which enables substantial improvements over ERM in all settings. Such flexibility and ease-of-implementation with respect to interventions of interest are key advantages of using T2I models for IDA.

## 4 ABLATION ANALYSIS

While in the previous sections we have experimented with various interventional prompting strategies using one of the simplest editing techniques (`SDEdit`), several other editing techniques exist and exhibit widely different behaviour. In this section, we study how the usage of different editing techniques may affect IDA performance: first, we investigate the use of alternative conditioning mechanisms that have been developed for T2I to study which are most effective Sec. 4.1; and second, we consider the earlier finding that filtering low-quality examples can improve synthetic data from earlier T2I generators (He et al., 2023) to determine whether this is also true of more recent, improved generators (Sec. 4.2).

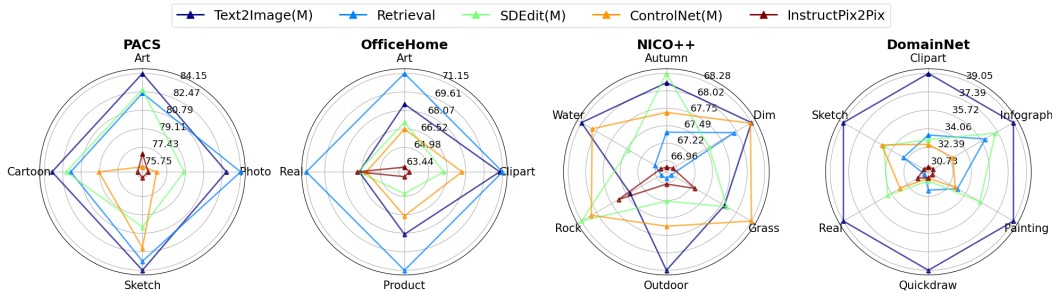

**Figure 5: SDG Results by Conditioning Mechanism.** Results are reported following the same format as Fig. 2.

## 4.1 DOES THE CHOICE OF THE EDITING TECHNIQUE MAKE A DIFFERENCE?

As we have seen, `SDEdit` is sufficient to simulate interventions that yield improved performance on the downstream tasks. However, there are cases in which such a simple editing technique is not sufficient to obtain the desired augmentation (e.g. see the second row of Fig. 4). This might depend on the fact `SDEdit` initialises the diffusion process from an embedding of the image being edited. This form of conditioning of the denoising process may be too constraining and inadequate to obtain the desired manipulation.[4] However, several other conditioning mechanisms exist. We consider three other forms of conditioning that may be suitable for our goal: `Text2Image`, `ControlNet` and `InstructPix2Pix`. With `Text2Image` we refer to the native ability of Stable Diffusion of generating images by only conditioning on the text: the diffusion process is initialised with random noise, and the prompt embeddings are used to condition the attention matrices in the denoising steps. In this case, the text is steering the diffusion in order to produce samples that approximate its training distribution. `ControlNet` (Zhang et al., 2023) aims at inducing stronger spatial consistency between the original and the augmented image by using an additional network that has been trained to condition the generative process on spatial guidance ("Canny edges"; Canny, 1986). `InstructPix2Pix` (Brooks et al., 2022) aims at improving the ability of diffusion models to follow editing instructions by training it on tuples of original images, editing instructions and desired editing outputs.

**Single Domain Generalization** In Fig. 5, we can see that the conditioning can have an extreme impact on the performance. First, we observe that in general `InstructPix2Pix` seems to underperform with respecct to other conditioning forms in most of the cases. This may be associated to the fact its training set (which is distilled from Stable Diffusion) may not contain enough variety of samples to provide an adequate implementation of a general interventional mechanism. Although `ControlNet` allows for a better spatial control, its performance is similar to or lower than `SDEdit` in most of the cases. This might be expected when considering that this evaluation task does not particularly benefit from the preservation of spatial features. More surprisingly, `Text2Image` can be an extremely effective conditioning technique. The success of this approach indicates that conditioning on an image may be a hindrance in order to approximate the desired domain.

**Reducing Reliance on Spurious Features** We observe that, although each conditioning technique helps in reducing the bias of the classifier, none of them is superior to all the others across all settings. This indicates the choice of conditioning can substantially affect the way the spurious features are perturbed. `Text2Image` is quite effective both at reducing the overreliance on the textures and background, resulting to be the most effective technique in these two settings. However, `ControlNet`'s ability to preserve the spatial features (i.e. the edges) of an image while modifying other aspects (in this case, the hair colour) yields superior performance, as the Canny edge detector is designed to omit a significant amount of information about the texture of objects, making editing it much easier. While `InstructPix2Pix` seems to be quite effective in removing overreliance on texture and background, it is not as effective as `ControlNet` on CelebA. `SDEdit` seems to be less effective than other techniques in reducing background bias, but is among the most effective when reducing texture and demographic biases.

**Retrieval Is Not (Always) Enough.** The strong performance we observe when removing source images from the generative process (i.e., substituting `SDEdit` for `Text2Image`) suggests that

---

[4]We have observed that this phenomenon persists across different values of the hyperparameter controlling the strength of conditioning.

Stable Diffusion's effectiveness may not be relying mostly on the input images. Therefore, we wonder whether we actually need to utilize Stable Diffusion as a generator at all: can we achieve similar results by simply using interventional prompts to retrieve relevant images from its original training dataset? To answer this question, we configure a retrieval baseline to compare the results of generating images and retrieving images from Stable Diffusion's training set using a simple image retrieval system,[5] querying it with the same minimal prompt that is used to generate images (see Fig. 11).

We observe large differences between the behaviors of the `Retrieval` method across the tasks we consider. In SDG, retrieval proves to be an extremely effective technique as shown in Fig. 5. For example, `Retrieval` outperforms all other methods on OfficeHome; and on PACS it proves to be only marginally inferior to `Text2Image(M)`. This is likely because Stable Diffusion's training data contains ample data from the classes and domains covered by these benchmarks and it is relatively easy to retrieve this data. On the other hand, for NICO++ and DomainNet, the retrieval baseline performance is inferior to `Text2Image(M)`. However, when Reducing Reliance on Spurious Features, `Retrieval` underperforms with respect to most generative techniques. This disagreement suggests that both retrieval and generative approaches are of interest and worth pursuing for different applications and different downstream tasks. Indeed, both have their own advantages and disadvantages.

In favor of retrieval, retrieved images do not generally contain unrealistic artifacts; and once the retrieval engine is deployed, it is significantly faster than generation. However, such deployment requires massive storage resources ($> 200$TB) and relies on highly efficient indexing and computing infrastructure. In contrast, generative models are significantly more compact in terms of storage (the version of Stable Diffusion we use is $\sim 8$GB) and do not require a dedicated infrastructure to be run. Perhaps more importantly, we find that modern generators can effectively produce samples that combine concepts from their training data in novel ways: in Appendix I.1, we compare the results of generating images with peculiar prompts versus retrieving them. Although the individual entities specified in the prompts are in the training set, we were unable to retrieve any images depicting the specific combination of entities and relations between them that was specified in the prompt, but Stable Diffusion is still able to reliably compose them (see Appendix I).[6]

### 4.2    DO WE STILL NEED POST-HOC FILTERING?

Although the quality of the generated samples of state-of-the-art diffusion models is impressive, failure cases may still occur and low-quality samples may be generated. Since such samples have been observed to harm the performance on downstream tasks, He et al. (2023) and Vendrow et al. (2023) deploy post-hoc filtering using CLIP (Radford et al., 2021) to discard them. In the case of IDA, the generated sample may fail at capturing either the specified class, the conditions of the environment we aim at simulating, or both. Therefore, we filter images that do not exhibit a high enough CLIP similarity score with respect to both prompts: one describing the class, the other describing the domain ("An image of a `class`" and "`domain`", respectively). Before training, we remove the samples with scores lower than a given percentile threshold, and provide our results in Fig. 8. Unlike He et al. (2023); Vendrow et al. (2023), we do not find that CLIP filtering yields consistent and substantial improvements. This may be due to the improved performance of newer generators or the fact that we are considering different tasks (SDG and RRSF). For further details, full results, and selected examples, see Appendix D.

## 5    CONCLUSION

In this work, we study how T2I generators can be used to perform interventional data augmentation in two settings, SDG and RRSF, finding they perform much better than traditional data augmentation techniques. We carry out a detailed investigation of how various components of the generative process may affect the results the most, concluding that the conditioning mechanism is the most important. Finally, we discuss the limitations of T2I-enabled interventional data augmentation in relation to the pros and cons of using retrieval for these tasks.

---

[5]Accessible at `https://rom1504.github.io/clip-retrieval`.

[6]However, these generators are not capable of synthesizing images depicting concepts that are completely absent or poorly described in their training data (e.g., medical images; see Appendix I.4), even when a few samples are used to learn a new corresponding token via textual inversion (Gal et al., 2022).

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

---

**Algorithm 1** Augmentation Algorithm

---

**Input:** Source Domain $\mathcal{D}_{\text{train}} = \{(\mathbf{x}_i, y_i)\}_{i=1}^N$, Target Domain Prior Knowledge $\mathcal{P}(\mathcal{C}_{\text{test}})$, PromptingStrategy. GenerativeModel, Model $f(\theta, \mathbf{x})$
**Output:** Trained $f(\cdot)$
    *Pre-generation Stage* :
 1: **for** $(\mathbf{x}_i, y_i)$ in $\mathcal{D}_{\text{train}}$ **do**
 2:    *# select **k** prompts for each original image*
 3:    $Prompts = []$
 4:    **for** _ in $range(k)$ **do**
 5:        $Prompts.append($PromptingStrategy$(y_i, \mathcal{P}(\mathcal{D}_{\text{test}})))$
 6:    **end for**
 7:    *# generate **one** sample for each prompt*
 8:    $[\hat{\mathbf{x}}_i] \leftarrow$ GenerativeModel$(Prompts , \mathbf{x}_i)$
 9:    Save augmented samples in $\mathcal{S} = \{\mathbf{x}_i : [\hat{\mathbf{x}}_i]\}_{i=1}^N$
10: **end for**
    *Training Stage* :
11: **for** Batch $(\mathbf{x}_i, y_i)_{i=1}^m$ in $\mathcal{D}_{\text{train}}$ **do**
12:    $(\mathbf{x}_i, \hat{\mathbf{x}}_i) \leftarrow$ Concat$(\mathbf{x}_i,$ RandomSelect$(\mathcal{S}[\mathbf{x}_i]))$
13:    TrainingStep$(\mathbf{x}_i, \hat{\mathbf{x}}_i)$
14: **end for**
15: **return** $f(\cdot)$

---

# A  EXPERIMENT IMPLEMENTATION

## A.1  GENERAL SETUP

Due to the speed limitation of generative models[7], we pre-generate all the augmentation images. For each image in the training set, we randomly selected $k$ text prompts from the templates (see Appendix J). In the Single Domain Generalization experiments, we choose $k = 3$ (for PACS and OfficeHome) and $k = 5$ (for NICO++ and DomainNet) prompts for each image (i.e., one prompt from each target domain), whereas for the weakening spurious correlation experiments, we choose $k = 4$ prompts for each image to randomize the correlation between the causal and spurious features. Then for each prompt **one** image will be generated and saved as a corresponding augmented version of the original image. At training time, for each training image in the batch, one of its augmented versions will be randomly selected from the $k$ pre-generated intervened samples. The general augmentation pipeline is shown in Algorithm 1.

On efficiency, we note that, given a dataset with $N$ traing samples, the generated interventional data will have a size of $N \times k$, where $k$ ranges from $3 \sim 5$ (depending on the experiment). As such, the number of generated samples is generally low (given $N$ is often of a few thousands) with respect to the amount of samples generated by baseline augmentation techniques (which is $N \times e$ where $e$ represents the number of training epochs, and typically[8] $e >> k$). Although baselines produce more augmentations of the same image, our technique requires fewer augmentations per image to attain superior performance as `SDEdit` can intentionally target specific types of interventions.

We report a few statistics about the training, validation and test set sizes as well as the number of classes for each dataset in Tab. 23. We use the model checkpoint of the last epoch to measure the test accuracy. For the experiments, as typical in the literature, we use pre-trained models on ImageNet for the backbones. To reproduce the experiment, we make part of our implementation available in the following anonymous repository.

## A.2  SINGLE DOMAIN GENERALIZATION

We set up the experiment under the standard Single Domain Generalization paradigm. For **PACS**, **Office-Home**, **NICO++**, and **DomainNet**, we train a model for each single domain and

---

[7]Significant progress in generation speed has been performed from the first versions of Stable Diffusion to the one we have been using in this paper. Accelerating diffusion models is an active area of research
[8]E.g., in our experiments, $e = 50$.

**Table 3:** Single Domain Generalization (SDG) PACS result with ResNet-18.

| | Art | Photo | Sketch | Cartoon | Average |
|---|---|---|---|---|---|
| ERM | 74.8 | 39.67 | 48.12 | 72.37 | 58.74 |
| MixUp | 67.14 | 39.57 | 33.24 | 63.27 | 50.81 |
| CutMix | 68.46 | 36.5 | 31.99 | 67.2 | 51.04 |
| AugMix | 68.88 | 38.75 | 43.89 | 76.86 | 57.09 |
| RandAugment | 69.07 | 44.48 | 49.36 | 72.31 | 58.8 |
| CutOut | 69.19 | 37.77 | 40.72 | 71.77 | 54.86 |
| RSC | 71.18 | 41.04 | 46.56 | 72.17 | 57.74 |
| MEADA | 70.32 | 39.55 | 44.94 | 74.03 | 57.21 |
| PixMix | 69.49 | 47.5 | 54.72 | 77.06 | 62.19 |
| L2D | 84.07 | 51.06 | 50.94 | 77.12 | 65.8 |
| ACVC | 72.65 | 43.33 | 60.35 | 78.98 | 63.83 |
| VQGAN-CLIP(M) | 78.09 | 54.38 | 53.78 | 77.76 | 66.00 |
| Retrieval | 81.22 | 75.49 | 83.36 | 83.24 | 80.83 |
| SDEdit(M) | 82.27 | 58.87 | 72.76 | 81.93 | 73.96 |
| SDEdit(H) | 84.23 | 61.7 | 70.31 | 82.74 | 74.75 |
| SDEdit(LEC) | 81.08 | 62.04 | 62.19 | 82.18 | 71.87 |
| SDEdit(LEM) | 83.21 | 58.74 | 66.45 | 82.37 | 72.69 |
| Text2Image(LEM) | 80.59 | 68.82 | 83.58 | 85.27 | 79.56 |
| Text2Image(M) | 83.17 | 71.31 | 87.42 | 87.12 | 82.26 |
| ControlNet(M) | 77.07 | 54.64 | 75.78 | 81.81 | 72.32 |
| Textual Inversion | 78.57 | 67.67 | 68.66 | 83.9 | 74.7 |
| InstructPix2Pix | 76.08 | 56.22 | 50.79 | 78.39 | 65.37 |

evaluate it on the remaining unseen domains to measure the test accuracy. For the first two datasets, we generate **three** augmented samples each one of them corresponds to one target domain. For the latter two, we similarly generate **five**. For all the datasets, we use an image size of $224 \times 224$. The full experiment results with expanded test accuracy one each test domain for PACS/OfficeHome/NICO++/DomainNet are shown in Tab. 3/Tab. 5/Tab. 7/ Tab. 9 for ResNet-18; Tab. 4/Tab. 6 / Tab. 8/Tab. 10 for ResNet-50. The visualized comparison between traditional data augmentation and generative model-based image editing as well as comparison among different types of conditional generation strategy is shown in Fig. 6 and Fig. 7, respectively. An overall comparison between different editing techniques is also presented in Tab. 11.

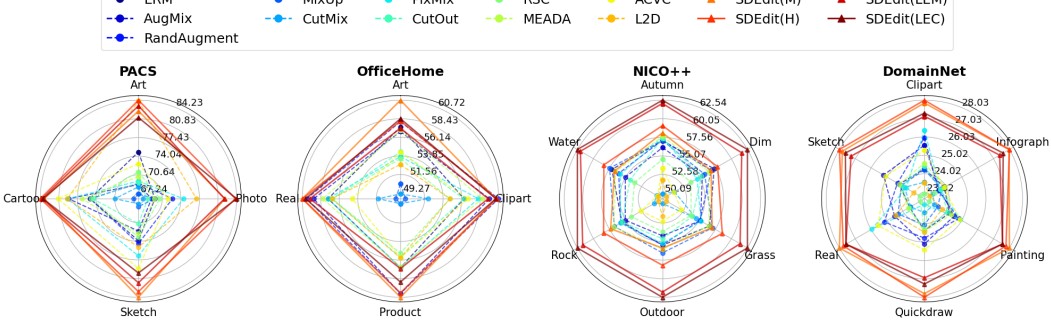

**Figure 6: Single Domain Generalization (SDG) Performance** results in comparison with baselines (dashed lines) and OURS (solid lines) using ResNet-18.

### A.3 SDG LARGE MODELS ABALATION

To show our method can scale to larger models, we perform an ablation study with ConvNeXt-L (Liu et al., 2022) (198M parameters) on PACS SDG experiment. As shown in Tab. 12 (average) Tab. 13 (detailed), we observe our method and still outperform all the baselines by a considerable margin. This proves the applicability of our method to different sizes of models.

**Table 4:** Single Domain Generalization (SDG) PACS result with ResNet-50. Columns are Single source domains; accuracies are the average test accuracy of the three remaining target domains when training using the indicated source domain (best accuracies are in bold).

| | Art | Photo | Sketch | Cartoon | Average |
|---|---|---|---|---|---|
| ERM | 74.44 | 48.78 | 50.89 | 73.74 | 61.96 |
| MixUp | 66.31 | 42.98 | 45.64 | 77.76 | 58.17 |
| CutMix | 72.53 | 40.03 | 44.72 | 76.72 | 58.5 |
| AugMix | 75.8 | 51.32 | 49.99 | 81.42 | 64.63 |
| RandAugment | 71.38 | 46.8 | 55.95 | 76.33 | 62.61 |
| CutOut | 76.67 | 42.69 | 48.93 | 75.2 | 60.87 |
| RSC | 73.15 | 53.47 | 51.11 | 80.58 | 64.58 |
| MEADA | 73.72 | 48.78 | 59.81 | 73.84 | 64.04 |
| PixMix | 77.33 | 55.58 | 52.42 | 83.15 | 67.12 |
| L2D | 77.33 | 58.41 | 58.14 | 81.7 | 68.89 |
| ACVC | 79.63 | 52.76 | 58.13 | 81.4 | 67.98 |
| SDEdit(M) × | 81.21 | 57.54 | 80.60 | 84.76 | 76.03 |
| SDEdit(O-M) | 82.59 | 65.44 | 79.3 | 83.64 | 77.74 |
| Retrieval | 82.36 | 76.24 | 87.0 | 86.01 | 82.9 |
| SDEdit(M) | 82.67 | 62.94 | 73.78 | 86.33 | 76.43 |
| SDEdit(H) | 83.68 | 64.22 | 78.95 | 84.63 | 77.87 |
| SDEdit(LEC) | 82.69 | 59.48 | 77.76 | 85.57 | 76.38 |
| SDEdit(LEM) | 84.11 | 59.1 | 73.39 | 86.0 | 75.65 |
| Text2Image(LEM) | 82.11 | 68.08 | 87.55 | 87.71 | 81.36 |
| Text2Image(M) | 84.15 | 72.9 | 90.51 | 87.34 | 83.72 |
| ControlNet(M) | 75.65 | 56.47 | 81.83 | 84.01 | 74.49 |
| Textual Inversion | 76.15 | 68.36 | 76.66 | 87.89 | 77.27 |
| InstructPix2Pix | 76.87 | 54.47 | 54.7 | 81.25 | 66.82 |

**Table 5:** SDG OfficeHome result with ResNet-18.

| | Art | Clipart | Product | Real | Average |
|---|---|---|---|---|---|
| ERM | 57.43 | 50.83 | 48.9 | 58.68 | 53.96 |
| MixUp | 50.41 | 43.19 | 41.24 | 51.89 | 46.68 |
| CutMix | 49.17 | 46.15 | 41.2 | 53.64 | 47.54 |
| AugMix | 56.86 | 54.12 | 52.02 | 60.12 | 55.78 |
| RandAugment | 58.07 | 55.32 | 52.02 | 60.82 | 56.56 |
| CutOut | 54.36 | 50.79 | 47.68 | 58.24 | 52.77 |
| RSC | 53.51 | 48.98 | 47.16 | 58.3 | 51.99 |
| MEADA | 57.0 | 53.2 | 48.81 | 59.21 | 54.55 |
| PixMix | 53.77 | 52.68 | 48.91 | 58.68 | 53.51 |
| L2D | 52.79 | 48.97 | 47.75 | 58.31 | 51.95 |
| ACVC | 54.3 | 51.32 | 47.69 | 56.25 | 52.39 |
| Retrieval | 65.02 | 63.55 | 60.51 | 64.32 | 63.35 |
| SDEdit(M) | 60.72 | 54.95 | 52.47 | 61.26 | 57.35 |
| SDEdit(H) | 58.15 | 55.12 | 51.94 | 61.24 | 56.61 |
| SDEdit(LEC) | 58.43 | 54.96 | 50.64 | 60.93 | 56.24 |
| SDEdit(LEM) | 57.27 | 53.97 | 49.02 | 60.5 | 55.19 |
| Text2Image(LEM) | 59.8 | 61.88 | 55.13 | 58.24 | 58.76 |
| Text2Image(M) | 62.77 | 64.57 | 57.51 | 61.2 | 61.51 |
| ControlNet(M) | 59.59 | 59.58 | 54.94 | 62.14 | 59.06 |
| InstructPix2Pix | 56.2 | 51.58 | 49.85 | 59.96 | 54.4 |

**Table 6:** SDG OfficeHome result with ResNet-50.

|  | Art | Clipart | Product | Real | Average |
|---|---|---|---|---|---|
| ERM | 63.62 | 61.32 | 56.85 | 65.99 | 61.94 |
| MixUp | 63.46 | 59.2 | 54.97 | 64.21 | 60.46 |
| CutMix | 59.3 | 54.45 | 51.9 | 63.0 | 57.16 |
| AugMix | 63.99 | 61.11 | 58.88 | 66.44 | 62.6 |
| RandAugment | 64.92 | 61.38 | 59.34 | 66.42 | 63.02 |
| CutOut | 62.15 | 58.24 | 55.77 | 63.98 | 60.03 |
| RSC | 60.91 | 56.86 | 54.21 | 64.41 | 59.1 |
| MEADA | 64.48 | 61.6 | 57.34 | 64.89 | 62.08 |
| PixMix | 63.54 | 60.34 | 57.29 | 64.54 | 61.43 |
| L2D | 60.79 | 55.01 | 54.76 | 62.93 | 58.37 |
| ACVC | 62.33 | 57.76 | 55.59 | 64.02 | 59.92 |
| Retrieval | 71.15 | 71.46 | 67.21 | 70.73 | 70.14 |
| SDEdit(M) | 67.08 | 64.48 | 60.01 | 67.06 | 64.66 |
| SDEdit(H) | 64.55 | 63.05 | 58.99 | 66.48 | 63.27 |
| SDEdit(LEC) | 65.96 | 63.12 | 58.6 | 66.03 | 63.43 |
| SDEdit(LEM) | 64.86 | 62.88 | 58.46 | 66.37 | 63.14 |
| Text2Image(LEM) | 65.72 | 68.8 | 62.61 | 64.09 | 65.31 |
| Text2Image(M) | 68.6 | 71.11 | 63.83 | 66.98 | 67.63 |
| ControlNet(M) | 66.52 | 66.65 | 62.12 | 66.47 | 65.44 |
| InstructPix2Pix | 63.34 | 60.39 | 58.43 | 67.13 | 62.32 |

**Table 7:** SDG NICO++ Result with ResNet-18.

|  | autumn | dim | grass | outdoor | rock | water | Average |
|---|---|---|---|---|---|---|---|
| ERM | 57.07 | 60.95 | 62.4 | 61.82 | 58.52 | 65.04 | 60.97 |
| RandAugment | 57.19 | 60.51 | 61.23 | 61.77 | 58.67 | 64.08 | 60.57 |
| AugMix | 56.19 | 59.18 | 61.29 | 60.72 | 58.1 | 63.16 | 59.77 |
| MixUp | 57.15 | 59.52 | 62.77 | 62.71 | 59.47 | 65.36 | 61.16 |
| CutOut | 57.42 | 59.07 | 60.33 | 61.07 | 58.48 | 62.5 | 59.81 |
| PixMix | 57.55 | 58.38 | 61.36 | 61.62 | 58.68 | 63.85 | 60.24 |
| RSC | 54.61 | 57.47 | 60.14 | 60.25 | 57.32 | 61.86 | 58.61 |
| ACVC | 53.43 | 54.91 | 58.94 | 59.07 | 56.11 | 58.67 | 56.85 |
| MEADA | 57.7 | 60.17 | 62.32 | 62.27 | 59.53 | 64.52 | 61.09 |
| L2D | 51.88 | 53.79 | 57.15 | 58.48 | 53.92 | 58.55 | 55.63 |
| Retrieval | 56.69 | 60.31 | 61.58 | 62.51 | 58.06 | 63.57 | 60.45 |
| SDEdit(M) | 58.17 | 60.48 | 62.72 | 62.16 | 59.95 | 64.66 | 61.36 |
| SDEdit(H) | 59.14 | 61.48 | 63.95 | 64.14 | 60.84 | 66.15 | 62.62 |
| SDEdit(LEC) | 62.54 | 65.97 | 67.01 | 67.85 | 64.15 | 69.85 | 66.23 |
| SDEdit(LEM) | 62.11 | 65.11 | 66.12 | 67.25 | 63.49 | 69.43 | 65.59 |
| Text2Image(M) | 58.89 | 63.79 | 63.56 | 64.85 | 58.9 | 66.3 | 62.72 |
| ControlNet(M) | 58.36 | 62.43 | 64.13 | 63.29 | 59.49 | 65.12 | 62.14 |
| InstructPix2Pix | 57.01 | 58.36 | 61.53 | 61.76 | 58.59 | 63.73 | 60.16 |

**Table 8:** SDG NICO++ Result with ResNet-50.

|  | autumn | dim | grass | outdoor | rock | water | Average |
|---|---|---|---|---|---|---|---|
| ERM | 66.74 | 70.37 | 72.05 | 71.3 | 66.58 | 72.64 | 69.95 |
| RandAugment | 67.23 | 71.43 | 70.81 | 70.62 | 66.47 | 72.71 | 69.88 |
| AugMix | 66.18 | 69.21 | 70.03 | 70.22 | 65.51 | 71.72 | 68.81 |
| MixUp | 67.6 | 70.3 | 72.47 | 72.26 | 67.12 | 74.01 | 70.63 |
| CutMix | 62.82 | 67.6 | 69.39 | 69.01 | 63.59 | 69.78 | 67.03 |
| CutOut | 66.76 | 69.34 | 70.13 | 70.13 | 66.67 | 72.33 | 69.23 |
| PixMix | 66.99 | 68.75 | 69.57 | 71.72 | 67.1 | 72.75 | 69.48 |
| RSC | 63.96 | 67.69 | 68.48 | 69.21 | 63.96 | 70.94 | 67.37 |
| ACVC | 63.74 | 67.48 | 67.73 | 68.71 | 63.89 | 69.95 | 66.92 |
| MEADA | 67.47 | 69.99 | 71.72 | 71.31 | 65.76 | 73.06 | 69.89 |
| L2D | 61.81 | 64.44 | 66.78 | 66.67 | 63.42 | 68.02 | 65.19 |
| SDEdit(M) × | 67.90 | 71.42 | 72.61 | 72.32 | 67.10 | 73.79 | 70.90 |
| Retrieval | 67.39 | 72.16 | 71.53 | 71.83 | 66.19 | 73.45 | 70.42 |
| SDEdit(M) | 68.28 | 71.42 | 72.68 | 72.31 | 67.95 | 74.07 | 71.12 |
| SDEdit(H) | 69.13 | 72.13 | 73.64 | 73.33 | 68.46 | 75.03 | 71.95 |
| SDEdit(LEC) | 70.21 | 74.68 | 75.05 | 75.54 | 69.74 | 76.89 | 73.69 |
| SDEdit(LEM) | 70.36 | 74.4 | 74.48 | 75.09 | 69.83 | 77.47 | 73.61 |
| Text2Image(M) | 68.14 | 72.67 | 72.63 | 73.77 | 66.88 | 75.17 | 71.54 |
| ControlNet(M) | 67.69 | 72.64 | 73.19 | 72.84 | 67.73 | 74.92 | 71.5 |
| InstructPix2Pix | 66.86 | 70.35 | 72.02 | 71.95 | 67.13 | 73.31 | 70.27 |

**Table 9:** SDG DomainNet Result with ResNet-18.

|  | clipart | infograph | painting | quickdraw | real | sketch | Average |
|---|---|---|---|---|---|---|---|
| ACVC | 25.31 | 19.86 | 25.23 | 8.0 | 27.49 | 26.84 | 22.12 |
| AugMix | 25.58 | 19.09 | 24.74 | 7.41 | 26.41 | 27.03 | 21.71 |
| CutMix | 23.56 | 17.83 | 23.0 | 4.33 | 25.36 | 25.04 | 19.85 |
| CutOut | 24.44 | 19.27 | 24.16 | 6.03 | 25.45 | 25.31 | 20.78 |
| ERM | 24.29 | 19.93 | 24.32 | 6.08 | 25.42 | 25.54 | 20.93 |
| L2D | 23.55 | 17.26 | 23.69 | 6.24 | 26.33 | 24.17 | 20.21 |
| MEADA | 24.6 | 20.06 | 24.5 | 6.17 | 25.52 | 25.56 | 21.07 |
| MixUp | 24.25 | 19.46 | 23.31 | 5.51 | 26.18 | 25.34 | 20.68 |
| PixMix | 26.39 | 19.18 | 25.28 | 3.49 | 27.9 | 24.89 | 21.19 |
| RSC | 22.92 | 18.21 | 22.52 | 6.11 | 24.72 | 23.59 | 19.68 |
| RandAugment | 25.99 | 18.88 | 25.12 | 6.83 | 27.08 | 25.71 | 21.60 |
| SDEdit(H) | 28.03 | 31.68 | 29.27 | 12.66 | 29.78 | 31.22 | 27.11 |
| SDEdit(LEC) | 27.33 | 30.56 | 28.96 | 11.35 | 29.6 | 30.65 | 26.41 |
| SDEdit(LEM) | 27.16 | 29.92 | 28.97 | 10.74 | 29.6 | 30.12 | 26.08 |
| SDEdit(M) | 27.88 | 31.62 | 29.57 | 12.3 | 30.03 | 30.94 | 27.06 |
| Text2Image(M) | 34.12 | 35.32 | 31.68 | 36.13 | 33.21 | 36.43 | 34.48 |
| ControlNet(M) | 28.19 | 23.40 | 27.59 | 18.81 | 29.28 | 31.62 | 26.48 |

**Table 10:** SDG DomainNet Result with ResNet-50.

|  | clipart | infograph | painting | quickdraw | real | sketch | Average |
|---|---|---|---|---|---|---|---|
| ACVC | 29.84 | 26.72 | 29.86 | 8.96 | 31.88 | 31.47 | 26.46 |
| AugMix | 30.04 | 26.1 | 29.48 | 8.92 | 31.07 | 31.61 | 26.20 |
| CutMix | 28.9 | 24.29 | 27.92 | 5.99 | 29.97 | 29.75 | 24.47 |
| CutOut | 28.98 | 25.8 | 28.71 | 6.6 | 29.96 | 29.36 | 24.90 |
| ERM | 29.06 | 27.07 | 28.87 | 6.92 | 29.85 | 29.8 | 25.26 |
| L2D | 28.15 | 23.85 | 28.61 | 7.12 | 31.25 | 29.53 | 24.75 |
| MEADA | 29.09 | 26.77 | 28.81 | 6.81 | 30.06 | 30.05 | 25.26 |
| MixUp | 29.34 | 26.89 | 29.17 | 6.46 | 30.66 | 30.42 | 25.50 |
| PixMix | 30.77 | 26.96 | 29.95 | 3.68 | 32.94 | 28.87 | 25.53 |
| RSC | 26.89 | 24.12 | 26.48 | 5.79 | 28.7 | 27.96 | 23.32 |
| RandAugment | 30.28 | 26.51 | 29.96 | 8.31 | 31.82 | 30.14 | 26.17 |
| SDEdit(H) | 32.89 | 37.95 | 33.99 | 15.89 | 34.45 | 35.77 | 31.82 |
| SDEdit(LEC) | 32.35 | 37.14 | 33.83 | 15.62 | 34.32 | 35.39 | 31.44 |
| SDEdit(LEM) | 31.84 | 36.75 | 33.72 | 13.88 | 34.19 | 35.24 | 30.94 |
| SDEdit(M) | 33.09 | 38.13 | 33.99 | 15.86 | 34.64 | 35.94 | 31.94 |
| InstructPix2Pix | 30.63 | 27.29 | 30.04 | 14.70 | 32.27 | 30.99 | 27.65 |
| ControlNet(M) | 32.66 | 30.72 | 31.93 | 14.92 | 33.66 | 36.15 | 30.01 |
| Text2Image(M) | 39.05 | 41.28 | 36.78 | 48.422 | 38.18 | 40.89 | 40.77 |

**Table 11:** Comparison Between Editing and Condition Strategies.

|  | PACS | OfficeHome | NICO | DomainNet |
|---|---|---|---|---|
| SDEdit(M) | 76.43 | 64.66 | 71.12 | 31.94 |
| Text2Image(M) | 83.72 | 67.63 | 71.54 | 40.77 |
| ControlNet(M) | 74.49 | 65.44 | 71.50 | 30.01 |
| InstructPix2Pix | 66.82 | 62.32 | 70.27 | 27.65 |
| Retrieval | 82.90 | 70.14 | 70.42 | 31.07 |

**Table 12:** SDG PACS result with ConvNeXt-L.

|  | Art | Photo | Sketch | Cartoon | Average |
|---|---|---|---|---|---|
| ERM | 79.8 | 60.62 | 50.76 | 86.53 | 69.43 |
| AugMix | 82.03 | 59.95 | 74.62 | 86.41 | 75.75 |
| RandAugment | 74.17 | 56.5 | 72.06 | 84.36 | 71.77 |
| MixUp | 79.82 | 60.74 | 55.45 | 84.35 | 70.09 |
| CutMix | 79.8 | 51.13 | 65.45 | 88.22 | 71.15 |
| CutOut | 81.41 | 55.55 | 71.8 | 85.16 | 73.48 |
| RSC | 84.26 | 55.3 | 76.15 | 86.13 | 75.46 |
| MEADA | 80.43 | 63.82 | 70.51 | 83.97 | 74.68 |
| ACVC | 83.68 | 68.86 | 77.99 | 85.03 | 78.89 |
| PixMix | 83.18 | 68.39 | 66.32 | 85.4 | 75.82 |
| L2D | 86.03 | 75.9 | 69.84 | 89.37 | 80.28 |
| SDEdit(M) | 86.53 | 74.5 | **89.22** | **88.98** | 84.81 |
| SDEdit(H) | **88.31** | **81.64** | 86.3 | 88.19 | **86.11** |
| SDEdit(LEC) | 87.7 | 75.05 | 85.77 | 87.04 | 83.89 |
| SDEdit(LEM) | 87.14 | 81.16 | 79.26 | 86.31 | 83.47 |

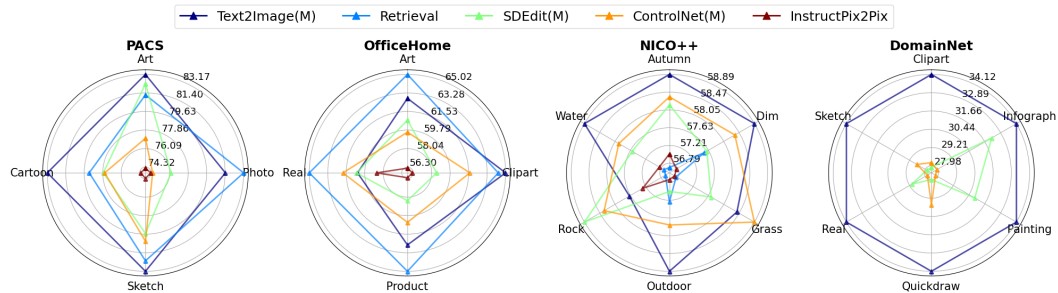

**Figure 7:** Comparison Between Different Condition Generation Strategy using ResNet-18.

**Table 13:** PACS result with ConvNeXt-L

| Source Domain | art | | | photo | | | sketch | | | cartoon | | |
|---|---|---|---|---|---|---|---|---|---|---|---|---|
| Target Domain | photo | sketch | cartoon | art | sketch | cartoon | art | photo | cartoon | art | photo | sketch |
| ERM | 98.86 | 71.93 | 68.6 | 75.78 | 51.39 | 54.69 | 40.82 | 53.65 | 57.81 | 89.75 | 93.65 | 76.2 |
| AugMix | 97.96 | 77.6 | 70.52 | 75.93 | 42.35 | 61.56 | 77.34 | 67.6 | 78.92 | 88.96 | 93.11 | 77.17 |
| RandAugment | 97.9 | 61.29 | 63.31 | 77.73 | 37.54 | 54.22 | 71.97 | 70.36 | 73.85 | 86.52 | 93.89 | 72.66 |
| MixUp | 99.04 | 68.57 | 71.84 | 81.1 | 43.62 | 57.51 | 49.46 | 54.91 | 61.99 | 85.35 | 95.93 | 71.77 |
| CutMix | 99.1 | 70.2 | 70.09 | 76.76 | 30.54 | 46.08 | 60.21 | 66.95 | 69.2 | 90.09 | 96.47 | 78.09 |
| CutOut | 98.62 | 74.32 | 71.29 | 76.32 | 39.09 | 51.24 | 72.51 | 70.36 | 72.53 | 86.62 | 93.11 | 75.74 |
| RSC | 98.98 | 79.54 | 74.27 | 76.61 | 40.75 | 48.55 | 69.34 | 80.96 | 78.16 | 85.35 | 93.77 | **79.28** |
| MEADA | 98.92 | 72.66 | 69.71 | 76.27 | 52.56 | 62.63 | 61.08 | 76.77 | 73.68 | 86.96 | 92.99 | 71.95 |
| ACVC | 98.08 | 80.53 | 72.44 | 81.84 | 65.44 | 59.3 | 83.69 | 74.01 | 76.28 | 87.3 | 92.93 | 74.85 |
| L2D | 98.98 | **83.1** | 76.02 | 80.86 | **80.02** | 66.81 | 68.6 | 68.86 | 72.06 | **91.55** | 97.13 | 79.43 |
| PixMix | **99.46** | 75.16 | 74.91 | 79.98 | 62.26 | 62.93 | 62.84 | 65.27 | 70.86 | 86.13 | 92.1 | 77.98 |
| SDEdit(M) | 99.34 | 76.43 | 83.83 | 82.96 | 64.04 | 76.49 | **87.79** | 92.28 | **87.59** | 91.06 | **97.19** | 78.7 |
| SDEdit(H) | 99.1 | 81.67 | **84.17** | 85.89 | 78.54 | 80.5 | 82.86 | 92.69 | 83.36 | 91.06 | 96.11 | 77.4 |
| SDEdit(LEC) | 98.26 | 82.59 | 82.25 | 85.64 | 67.96 | 71.54 | 85.4 | **92.93** | 78.97 | 87.94 | 94.61 | 78.57 |
| SDEdit(LEM) | 99.4 | 79.33 | 82.68 | **86.77** | 74.42 | **82.3** | 77.64 | 84.19 | 75.94 | 87.35 | 95.09 | 76.48 |

## A.4 Effect of Accessing Multiple Source Domain

We also present further investigation on the effect of accessing multiple source domains, potentially including the target test domain. We experimented on three settings: (a) MDG: classifier trained on all but the target domain, which is the standard set-up for multi-domain generalization, where multiple domains of source data are used for training and one unseen domain is used for testing; (b) All: classifier trained on all the domains, (c) Target: classifier trained only on the target domain. As shown in Tab. 14, While ERM(Target/All) achieves almost perfect performance, this is expected as the training source domain includes the target test domain. Note that the accuracy in the table below is not directly comparable to SDG in the main paper since our main setting is Single Domain Generalization (SDG), where we have a single source domain for training, and the accuracy reported is the average test accuracy on multiple unseen test domains. However, here we have a single unseen target domain for testing. To provide a direct comparison between ERM and SDEdit, we experiment with SDEdit under the same setting in MDG with a minimal prompt. We demonstrate under MDG setting SDEdit also leads to significant performance improvement in all unseen test domains.

**Table 14:** Impact of Assessing Multiple Real/Synthetic Domain.

| | Art | Photo | Sketch | Cartoon | Average |
|---|---|---|---|---|---|
| ERM (Target) | 99.65 | 99.94 | 99.64 | 99.66 | 99.72 |
| ERM (All) | 99.71 | 99.70 | 99.84 | 99.60 | 99.74 |
| ERM (MDG) | 80.01 | 96.28 | 73.86 | 76.28 | 81.61 |
| OURS(MDG) | 87.5 | 95.75 | 79.21 | 85.2 | 86.91 |

**Table 15:** ImageNet-9 result with ResNet-18

|  | I.I.D. Test | Mixed Rand | Mixed Same | **Gap** ($\downarrow$) |
|---|---|---|---|---|
| ERM | 95.16 | 73.54 | 86.02 | 12.48 |
| MixUp | 94.62 | 67.63 | 83.91 | 16.28 |
| CutMix | 95.36 | 65.21 | 84.77 | 19.56 |
| AugMix | 95.16 | 74.73 | 87.72 | 12.99 |
| RandAugment | 96.69 | 78.20 | 90.44 | 12.25 |
| CutOut | 95.46 | 71.10 | 85.41 | 14.31 |
| RSC | 94.12 | 74.72 | 84.39 | 9.68 |
| MEADA | 95.56 | 74.74 | 87.43 | 12.69 |
| PixMix | 97.04 | 79.76 | 91.96 | 12.20 |
| ACVC | 93.97 | 76.38 | 88.16 | 11.77 |
| L2D | 92.84 | 73.04 | 84.10 | 11.06 |
| SDEdit(H) | 91.85 | 73.33 | 82.96 | 9.63 |
| Text2Image(H) | 90.12 | 69.63 | 75.8 | 6.17 |
| ControlNet(H) | 91.85 | 75.19 | 85.68 | 10.49 |
| InstructPix2Pix | 92.84 | 78.89 | 88.15 | 9.26 |
| Retrieval | 91.6 | 73.83 | 80.12 | 6.29 |

**Table 16:** Texture result with ResNet-18

|  | I.I.D. Test | Random | **Texture Bias** ($\downarrow$) |
|---|---|---|---|
| ERM | 81.75 | 18.77 | 72.45 |
| MixUp | 77.36 | 19.23 | 71.69 |
| CutMix | 79.96 | 15.64 | 77.16 |
| AugMix | 82.2 | 20.08 | 72.42 |
| RandAugment | 83.09 | 18.9 | 72.2 |
| CutOut | 81.85 | 17.81 | 74.19 |
| RSC | 79.9 | 20.48 | 71.11 |
| MEADA | 81.97 | 19.5 | 71.14 |
| PixMix | 80.91 | 26.86 | 64.64 |
| ACVC | 81.13 | 29.33 | 59.25 |
| L2D | 80.06 | 23.55 | 62.12 |
| SDEdit(H) | 85.94 | 31.48 | 55.46 |
| Text2Image(H) | 86.44 | 35.23 | 51.21 |
| ControlNet(H) | 84.13 | 21.88 | 62.58 |
| InstructPix2Pix | 79.75 | 26.17 | 53.58 |
| Retrieval | 85.85 | 33.91 | 51.94 |

# B  WEAKEN SPURIOUS CORRELATION

In the three considered cases, the reliance on the spurious correlation is measured as: (1) **ImageNet-9** (Background Bias): **Gap**, as defined in (Xiao et al., 2021), is the difference between the accuracies measured on the test sets `mixed same` and `mixed rand`. (2) **CCS Dataset** (Texture Bias): **Texture Bias**, as defined in (Geirhos et al., 2019), is the number of correct texture classifications over sum of the true positive texture and shape classifications. In the test CCS dataset, each image is synthesized with a texture and subject from different classes (i.e texture: elephant, class: cat). The true positive texture classification is the percentage of cases in which the model predicts the texture label correctly; similarly, true positive shape classification is the percentage of correctly classified shape labels. (3) **CelebA-sub** (Demographic Bias): **RandGap** and **FlipGap** represent the accuracy gap between **I.I.D** distribution to `rand` and `flip` respectively. The purpose is to measure the reliance on the spurious feature both for the average case (i.e., randomizing the spurious correlation in the test set) and in the worst case (i.e., the test set flips the spurious correlation). For all the three dataset each original image sample will have **four** pre-generated augmented samples. The comparison with all the baselines is in with ResNet-18 Tab. 15, Tab. 17, and Tab. 16.

## B.1  ADDITIONAL EXPERIMENT ON CIFAR-10-C

**Table 17:** CelebA-sub result with ResNet-18

|  | I.I.D. Test | Flip | Random | **FlipGap** ($\downarrow$) | **RandGap** ($\downarrow$) |
|---|---|---|---|---|---|
| ERM | 99.44 | 77.16 | 88.48 | 22.28 | 11.32 |
| MixUp | 99.16 | 79.4 | 88.86 | 19.76 | 9.46 |
| CutMix | 99.24 | 74.82 | 86.92 | 24.42 | 12.1 |
| AugMix | 99.56 | 76.42 | 87.82 | 23.14 | 11.4 |
| RandAugment | 99.04 | 77.62 | 88.96 | 21.42 | 11.34 |
| CutOut | 99.48 | 78.24 | 89.72 | 21.24 | 11.48 |
| RSC | 99.52 | 81.7 | 91.8 | 17.82 | 10.1 |
| MEADA | 99.48 | 77.24 | 89.08 | 22.24 | 11.84 |
| ACVC | 99.16 | 79.5 | 89.58 | 19.66 | 10.08 |
| PixMix | 99.32 | 76.62 | 88.34 | 22.7 | 11.72 |
| L2D | 99.12 | 81.96 | 90.96 | 17.16 | 9.0 |
| Retrieval | 98.6 | 77.9 | 89.3 | 20.7 | 11.4 |
| SDEdit(H) | 99.2 | 86.6 | 92.5 | 12.6 | 5.9 |
| Text2Image(H) | 98.8 | 90.0 | 93.6 | 8.8 | 3.6 |
| ControlNet(H) | 99.2 | 89.3 | 93.5 | 9.9 | 4.2 |
| InstructPix2Pix | 99.2 | 86.9 | 93.5 | 12.3 | 6.6 |

We conduct further experiments with the Cifar-10-C dataset. We adopt a similar setting as RRSF, where we train on Cifar-10 and test on Cifar-10-C. In the evaluation, domain shifts were organized into distinct groups for clarity. The following classifications were made:

- Blurring Effects: defocus blur, gaussian blur, glass blur, motion blur, zoom blur

- Noise Variations: gaussian noise, impulse noise, shot noise, speckle noise

- Compression Artifacts: JPEG compression

- Image Transformations: brightness, contrast, elastic transform, pixelate, saturate

|  | Blurring Avg. | Noise Avg. | Compression Avg. | Transformations Avg. | Overall Avg. |
|---|---|---|---|---|---|
| ERM | 72.04 | 74.01 | 75.71 | 73.93 | 73.55 |
| MixUp | 73.79 | 76.22 | 77.46 | 75.72 | 75.48 |
| PixMix | 75.84 | 79.41 | 81.39 | 79.32 | 78.63 |
| SDEdit(M) | 74.28 | 75.71 | 77.10 | 75.02 | 75.11 |

**Table 18:** Average performance of algorithms across grouped domain shifts.

As shown in Tab. 18, SDEdit still demonstrate effectiveness under various parametric domain shift. Although the generalization performance is inferior to other parametric augmentation methods, it can be used in a combined manner.

## B.2 HYPERPARAMETERS

**Training Hyperparameter** For all the other baselines, we use the value as proposed in their original papers or official implementation.

**Table 19:** Training Hyperparameters

|  | PACS | OfficeHome | NICO++ | ImageNet-9 | Texture | CelebA-sub |
|---|---|---|---|---|---|---|
| Epoch | 50 | 50 | 50 | 30 | 30 | 30 |
| Batch size | 64 | 64 | 64 | 64 | 64 | 64 |
| Warmup Epoch | 5 | 5 | 5 | 5 | 5 | 5 |
| Warmup Type | sigmoid | sigmoid | sigmoid | sigmoid | sigmoid | sigmoid |
| Weight Decay | 5e-4 | 5e-4 | 5e-4 | 5e-4 | 5e-4 | 5e-4 |
| Nesterov | True | True | True | True | True | True |
| Learning rate | 1e-3 | 3e-3 | 3e-3 | 1e-3 | 1e-3 | 1e-3 |
| Scheduler | Step | Step | Step | Step | Step | Step |
| Decay Step | 45 | 45 | 45 | 27 | 27 | 27 |
| Learning rate decay | 0.1 | 0.3 | 0.3 | 0.1 | 0.1 | 0.1 |

**Table 20:** Generator hyperparameters for each dataset

|                | PACS  | OfficeHome | NICO++ | DomainNet | ImageNet-9 | CelebA | Texture |
|----------------|-------|------------|--------|-----------|------------|--------|---------|
| Inference Step | 30    | 30         | 30     | 30        | 30         | 30     | 30      |
| Image Strength | 0.75  | 0.75       | 0.75   | 0.75      | 0.75       | 0.75   | 0.75    |
| Guidance Scale | 2.0   | 2.0        | 2.0    | 2.0       | 2.0        | 2.0    | 2.0     |
| Sampler        | UniPC | UniPC      | UniPC  | UniPC     | UniPC      | UniPC  | UniPC   |

**Generator Hyperparameter** For the two types of generative models, we use the hyperparameters for each dataset as shown in Tab. 20. Hyperparameters are tuned based on human judgement of few-shot image manipulation quality, without downstream task accuracy-based evaluation. Unspecified hyperparameters are set to their default value. For Stable Diffusion, We use "Runmyml/stable-diffusion-v1-5" pre-trained model. The training hyperparameters for setting are specified as shown in Tab. 19

## C  PROMPTING STRATEGIES

Hwere we detail how the prompts were obtained.

- *Domain expert (H)*: a collection of 1-8 simple "handcrafted" prompts per image domain (e.g., "an ink pen sketch of a(n) class"), authored by a human given only the domain descriptions provided by the respective benchmarks, without looking at any samples from the target domain.
- *Language enhancement (LE)*: following (He et al., 2023), we use the T5 language model (Raffel et al., 2020) fine-tuned on CommonGen (Lin et al., 2020)[9] to generate 1-8 prompts using only the domain and class labels as inputs. Two strategies, Conservative ($LE_C$) and Moderate ($LE_M$), are used: $LE_C$ deterministically generates consistent, high-probability outputs; and $LE_M$ is built to balance prompt diversity with quality. For both strategies, we use a T5 (Raffel et al., 2020) model that is pre-trained on both unsupervised language modeling of web text and supervised text-to-text language modeling tasks[10], then fine-tuned on CommonGen[11] (Lin et al., 2020). (We refer to this model as $T5_{CG}$.) CommonGen is a constrained-generation task whose objective is to generate a sentence describing a commonplace scenario that contains all words[12] provided in an input word set. For example, given the words {dog, frisbee, catch, throw}, an acceptable output is "The dog catches the frisbee when the boy throws it." (Lin et al., 2020) We always provide $T5_{CG}$ with a text input containing only a domain label and class label; for example, given a PACS image with domain sketch and class elephant, we simply feed $T5_{CG}$ "sketch elephant" as input. For $n$ number of prompts we will use to generate images, in $LE_C$, we simply use beam search decoding to generate prompts with $4n$ beams and select the top-$n$ highest probability beams. In $LE_M$, we use a conjunction of top-$k$ and top-$p$ (nucleus) sampling, with $k = 50$ and $p = 0.95$, returning $n$ sampled prompts. We experimented with other decoding configurations, but found that increasing prompt diversity (e.g., by increasing $k$, lowering $p$, or increasing temperature) consistently came at the cost of prompt quality.
- *Textual Inversion* (Gal et al., 2022): Given a set of images that share a common feature (e.g., belonging to the same class), this method learns an embedding in the text space that represents that feature. This embedding can be used to condition the generative process, thereby enhancing the generator's capability to reproduce that feature. Due to the computational cost associated with the additional training phase required by this approach, we limit its application to PACS. As shown in Tab. 3 and Tab. 4, Texual Inversion achieves 74.70% and 77.27% average accuracy for SDG. While outperforming all baseline methods, it is inferior to other relatively low-cost generative model-based strategies.

---

[9]https://huggingface.co/mrm8488/t5-base-finetuned-common_gen

[10]Pre-trained model (not directly used in experiments): https://huggingface.co/t5-base

[11]Fine-tuned model used in experiments: https://huggingface.co/mrm8488/t5-base-finetuned-common_gen

[12]Synonyms and inflected forms are also allowed (e.g., given input "eat", outputs containing "consume" or "eaten" are valid).

In order to yield the best IDA performance from a given T2I model, future work might consider strategies for directly optimizing prompts or utilizing human-in-the-loop prompt "debugging", as we discuss in Appendices E and F (respectively).

## D  CLIP FILTERING DETAILS

For each image in the generated dataset, we compute its CLIP similarity with respect to both prompts. Since the distributions of similarity scores can differ in scale and location, we cannot simply average the two scores in order to quantify how well a sample represents a class and a domain. Therefore, we sort the scores to produce two rankings and associate each image to the average of the percentile rank with respect to both prompts. We then discard a fixed amount of images with the lowest average percentile rank. (See Appendix D.1 for an example of top- and bottom-ranked images.) After filtering out the worst 10%, 25%, or 50% of synthetic images, we train our classifier on the remaining data. The results are displayed in Fig. 8. We find filtering to not yield consistent improvements across all the considered cases.

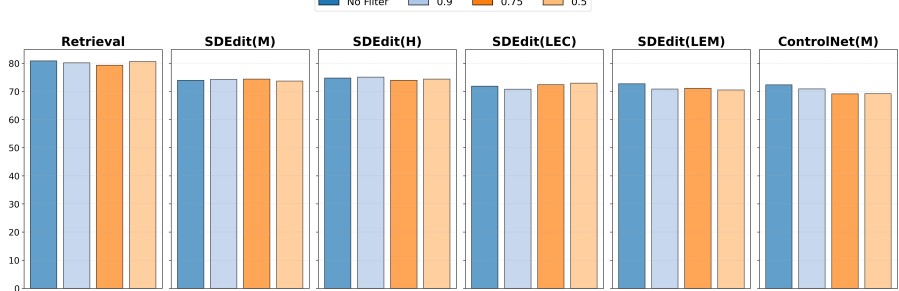

**Figure 8: CLIP Filtering Results.** SDG accuracies averaged across all test domains for different conditioning strategies (boxes in bold) and CLIP filtering proportions (colors).

### D.1  CLIP FILTERING EXAMPLES

Fig. 9 displays the best-matching (top) and worst-matching (bottom) synthetic images generated with SDEdit using LE$_M$ of class dog and domain cartoon according to their average percentile rank of CLIP similarity scores with the prompts "an image of a dog" and "a cartoon". In general, we observe that the images on the top do indeed appear to be cartoons and contain dogs (if somewhat disfigured in a few cases); whereas it seems that most of the images on the bottom either resemble *photos* of dogs (images 2, 4, 5, 6, and 8) or cartoons (images 1, 3, and 7), but do not generally seem to match both the target domain and the correct class.

## E  FULLY AUTOMATED APPLICATIONS

We examine a basic implementation of a fully-automated augmentation pipeline in the language enhancement (LE) experiments described above, finding that it sometimes achieved performance on-par with or exceeding that of the expert-handcrafted prompts. However, this language model is optimized to generate simple sentences describing commonplace scenarios (see Appendix C),

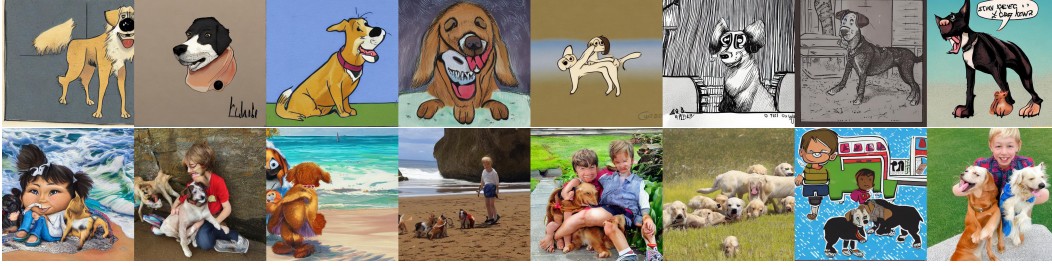

**Figure 9: CLIP Filtering Examples.** The most-similar (top) and least-similar (bottom) eight images according to their average percentile rank of CLIP similarity scores computed with respect to the provided prompts.

not image-generation prompts. Thus, it is possible that fine-tuning language models to generate prompts that are better optimized for downstream T2I generators may yield superior results to expert-handcrafted prompts in many scenarios, making this approach a promising direction for future work. Another approach to improve fully-automated prompting involves continuous prompt optimization (also known as "prompt tuning" or "soft prompting"). Recently, these methods have been shown to outperform human-interpretable prompts for a variety of natural-language (Li & Liang, 2021; Liu et al., 2021; Min et al., 2022; Khashabi et al., 2022) and vision-and-language (Gal et al., 2022; Zhou et al., 2022) tasks. These methods are not directly applicable to domain generalization because they require labelled samples to learn continuous prompts; but we suggest that they may be a promising fully-automated approach to domain adaptation tasks[13] where prompt interpretability is not necessary (cf. (Khashabi et al., 2022)).

## F    HUMAN-IN-THE-LOOP APPLICATIONS

### F.1    PROMPT INTERPRETABILITY ENABLES HUMAN-IN-THE-LOOP DEBUGGING

Specifying interventions with natural language makes it possible to flexibly specify the type of manipulations desired. In the future, we expect practitioners could iteratively improve the collection of prompts to achieve improved performance.

We begin with a small set of handcrafted prompts (the ones used for the results reported in the main paper) and observe a decrease in texture bias of **5.44%**. Reasoning on the task at hand and the desired effect of the augmentations, we expand the prompts set to cover a broader range of textures to further decrease texture bias by an additional **2.84%** (see Appendix J and Tab. 16).

More generally, it is possible to "debug" augmentations by directly analyzing prompts and modifying them to better reflect the desired intervention (which is possible with zero exposure to the target domain, or before augmented images are even generated). For example, the top prompts generated by **LE$_M$** for OfficeHome's `art` domain and `computer` class include "art on a computer", "a man is working on a computer with a piece of art on it", etc., indicating that **LE$_M$** generated prompts describing scenarios where both the class and domain label refer to individual objects in a visual scene. In Appendix F.1, we describe a few simple steps that can automatically filter out many such prompts[14], illustrating the flexibility of the natural-language augmentation interface.

LE$_M$ is prone to generating prompts that treat OfficeHome (Venkateswara et al., 2017) domain labels as objects, not as visual domains or styles. Fortunately, the interpretability of natural-language prompts that makes it possible for us to diagnose this problem also enables us to filter out many such prompts. One approach is to map domain labels to the visual conditions they denote: for example, the `Product` label may be replaced with "white background", `Real World` with "photograph", etc. However, this solution requires some knowledge about test domains, which may not always be available. Alternatively, image-related keywords like "image", "depict", or "style" can be included in the input issued to T5$_{CG}$, and outputs which do not place these additional terms in the same minimal noun phrase as the domain label can be removed (e.g., "an artistic depiction of a computer" or "a product image of a candle" would be kept, whereas "art depicted on a computer" or "a product and an image of a candle" would be excluded)[15]. While both of these strategies require limited human oversight to successfully "debug" prompts, more sophisticated fully-automated augmentation pipelines might learn to make such changes on their own, e.g., by integrating downstream image classifier accuracy as feedback to fine-tune prompt-generation models.

### F.2    OTHER HUMAN-IN-THE-LOOP APPLICATIONS

The usage of T2I generators to approximate interventions facilitates a variety of novel use cases. For example, consider a "human-in-the-loop" (HITL) application context, where humans are available to provide interactive feedback to a model. In the HITL *active learning* paradigm, human

---

[13]I.e., where an unlabeled sample of the target domain is available to facilitate learning of the environmental features of the target domain (Ghifary et al., 2016).

[14]However, as our LE experiments are explicitly intended to operate fully autonomously (i.e., with no human intervention or supervision), we do not carry out a full-scale "debugged" version of this experiment – all reported OfficeHome results are from the "buggy" prompts.

[15]For clarity, T5$_{CG}$ can replace input terms with inflected forms in generated prompts, e.g., allowing input terms "art" and "depict" to occur as "artistic" and "depiction" (respectively) in outputs (see **??**).

**Table 21:** Quantitative Comparison on Computation Time

| | ERM | AugMix | RandAugment | MixUp | CutMix | RSC | L2D | ACVC | MEADA | OURS (online) |
|---|---|---|---|---|---|---|---|---|---|---|
| Time (s) | 14.2 | 33.1 | 42.7 | 27.3 | 28.4 | 18.0 | 41.1 | 127.8 | 92.2 | 21.2 |

experts perform the role of "oracles" that a model may "query" to provide labels of highly uncertain or novel inputs (Mosqueira-Rey et al., 2022). In contrast, the "human-in-the-loop debugging" paradigm elaborated above implements the *interactive machine teaching* paradigm (Ramos et al., 2020), which treats human collaborators as *teachers* that may provide interactive feedback to update the "curriculum"[16] of images used to train a model. For example, a human collaborator may observe that a model tends to perform poorly in the context of a given target domain, or that generated images do not capture some important stylistic properties of the domain. In response, they may easily compose or revise image-generation prompts with explicit reference to important features of the target domain. Critically, our approach allows human teachers to directly update visual curricula using natural language, providing models with feedback in much the same terms as one would a human student. We believe that the intuitiveness and efficiency of this approach makes it a promising approach to domain generalization, shifting the burden of the problem from human domain knowledge to natural language and thus enabling human collaborators to interactively instruct models without prerequisite domain expertise.

In particular, we argue that this benefit is particularly salient in the context of test-driven software engineering practice. Rather than blindly assuming that the performance on application-independent benchmarks will transfer to application-specific cases, engineers need to extensively document (often through natural language) the potential use cases and test conditions. The ability to directly specify these criteria via natural-language augmentations, or even directly reuse the documentation to generate training data, could be invaluable for controlling, predicting, and understanding the behavior of vision models in real-world applications.

## G  COMPUTATIONAL EXPENSE

Although the inference speed of generative models has greatly improved over time, we found that SD is still too slow to generate synthetic data on-the-fly during training, so we pre-generate and store augmented data to amortise the generation cost when experimenting on different architectures and training procedures. For each sample in $\mathcal{D}^S$, we randomly selected $k$ text prompts, and for each prompt, **one** augmented image was generated and stored. At training time, for each training image in the batch, one of its augmented versions will be randomly selected from the $k$ pre-generated intervened samples. The general statistics of computational expense of each type of generative model on an NVIDIA A40 GPU and generator with hyperparameters specified for OfficeHome experiment are as follows: Stable Diffusion 1.5 took up $\sim 8$GB of VRAM (for inference – we do not compute gradients for any experiments) and required $\sim 0.5$ seconds per sample generated on average. In addition to our qualitative assessments, we have conducted a quantitative comparison of various data generation methods, focusing on the time efficiency aspect. Specifically, we measured the time required to complete an epoch on the PACS dataset using a ResNet18 model. The results, detailed in Tab. 21, reveal that the online augmentation speed of our method is on par with other parametric data augmentation methods and notably faster than learning-based methods. It's important to note that while the offline generation time for our method is approximately 4 hours on a single A40 GPU, this process is a one-time requirement and can be performed offline. Consequently, once the data is generated, it can be reused multiple times, thereby offsetting the initial time investment.

## H  FURTHER EXPERIMENTS ON GENERATIVE MODELS

### H.1  MANIPULATING ONLY THE ENVIRONMENTAL FEATURES IS IMPORTANT

It is important to observe that the T2I generator can manipulate not only the environmental features but also the class-related ones. When the manipulated class-related features still resemble those of the original training set, the issue is alleviated. However, it is important for future generators to allow stronger control over which features are manipulated and which not through language. In some cases, a potential solution could be to provide a mask that indicates which are the environmental

---

[16]Note that, in our case, a curriculum is defined in terms of the domains from which training examples are drawn, not the order in which they are presented (cf. (Bengio et al., 2009)).

**Table 22:** Inpainting Result on ImageNet-9

|          | in    | mixed rand | mixed same | gap   |
|----------|-------|------------|------------|-------|
| ERM      | 95.06 | 71.85      | 83.58      | 11.73 |
| SDEdit(H) | 95.06 | 77.65     | 85.8       | 8.15  |
| Inpaint(H) | **96.05** | **80.62** | **87.16** | **6.54** |

**Table 23:** Dataset Statistics

|           | No.train | No.validation | No.test | No.classes |
|-----------|----------|---------------|---------|------------|
| PACS      | 8977     | 1014          | 9991    | 7          |
| Officehome | 14032   | 1556          | 15588   | 65         |
| NICO++    | 61289    | 7661          | 15322   | 60         |
| DomainNet | 410657   | 18000         | 157918  | 345        |
| ImageNet-9 | 2835    | 405           | 810     | 9          |
| Texture   | 9600     | 1600          | 1280    | 16         |
| CelebA-sub | 5000    | 500           | 1000    | 2          |

features to be manipulated. To exemplify the importance of controlling mainly the environmental variables, we show that, when the inpainting capabilities of Stable Diffusion can leverage ground-truth background masks to preserve the foreground area, this further improves the performance of our method on ImageNet-9 as shown in Tab. 22

# I  AUGMENTATION SAMPLES

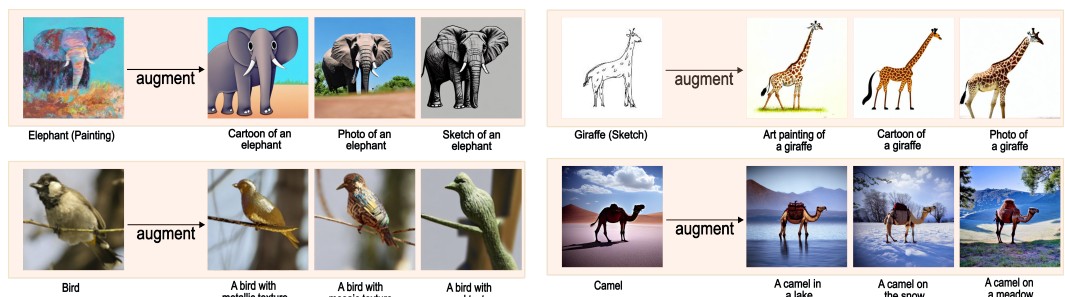

**Figure 10:** Interventional samples generated by Stable Diffusion. For each group of four images, the leftmost image is the original image, and the three images on the right are augmented samples with text prompts indicated.

## I.1  WHAT KIND OF INTERVENTIONS CAN THE GENERATOR APPROXIMATE?

In our experiments, we have shown that the way current T2I generators approximate interventions is sufficient to achieve good performance on standard benchmarks. The way T2I generators learn to approximate such manipulations is by leveraging large amounts of weakly-supervised data. Stable Diffusion trains on text-image pairs scraped from the web with minimal post-processing (weak supervision): this is significantly less expensive than manually providing class and domain labels (with the added effort of controlling the environmental conditions). A natural question is then whether generators can approximate forms of interventions that are not represented in the training set. This would require them to combine learned concepts in novel ways. We answer this question through a simple experiment: we compare the results of generating images and retrieving images from the training set through a search engine[17] (see Fig. 11). Although the individual entities specified in the prompts are in the training set, we were unable to retrieve any images depicting the specific combination of entities and relations between them that was specified in the prompt. Since the dataset we are querying is huge ($> 200$TB, which can be impossible to store in lack of extremely expensive hardware), it is infeasible to give a certain answer about whether a sample representing the query is present or not in it. Additionally, the system leverages CLIP embeddings to search for images

---

[17] Search Engine can be accessed through `https://rom1504.github.io/clip-retrieval`

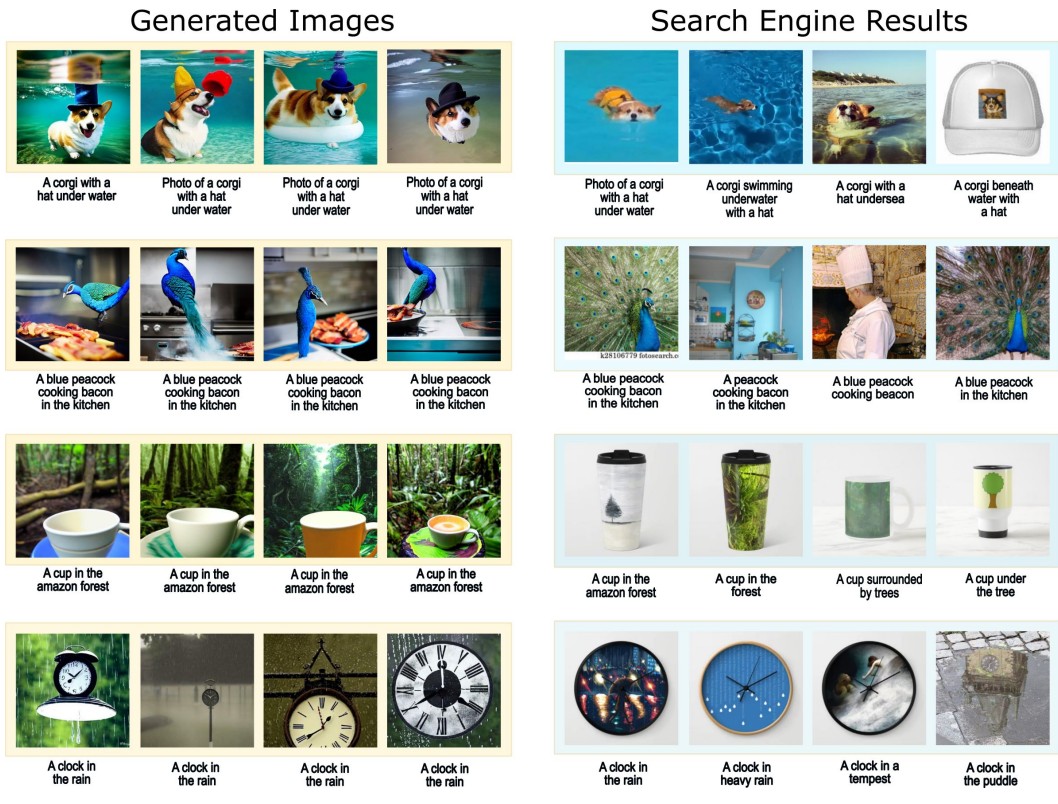

**Figure 11:** Comparison between Search Engine retrieval result and Stable Diffusion manipulation results. Images on the left are generated with Stable Diffusion; images on the right are retrieved from LAION-5B by querying the search engine with the prompt indicated below

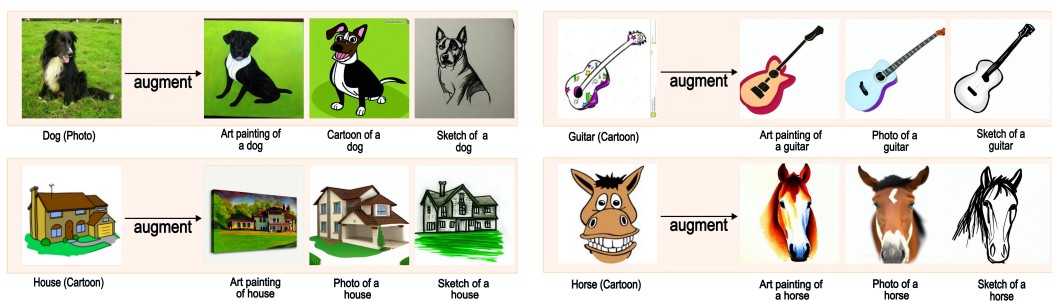

**Figure 12:** Stable Diffusion manipulation for in-distribution samples with prompt indicated below. For each group of images of four, the first image on the left is the original image, and the rest three are manipulated images

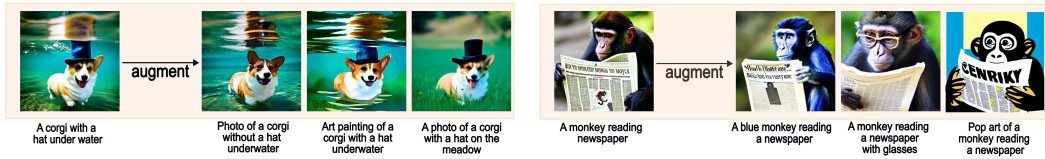

**Figure 13:** Stable Diffusion manipulation out-distribution samples with prompt indicated below. For each group of images of four, the first image is generated with prompt indicated from scratch, and the rest three are manipulated base on that.

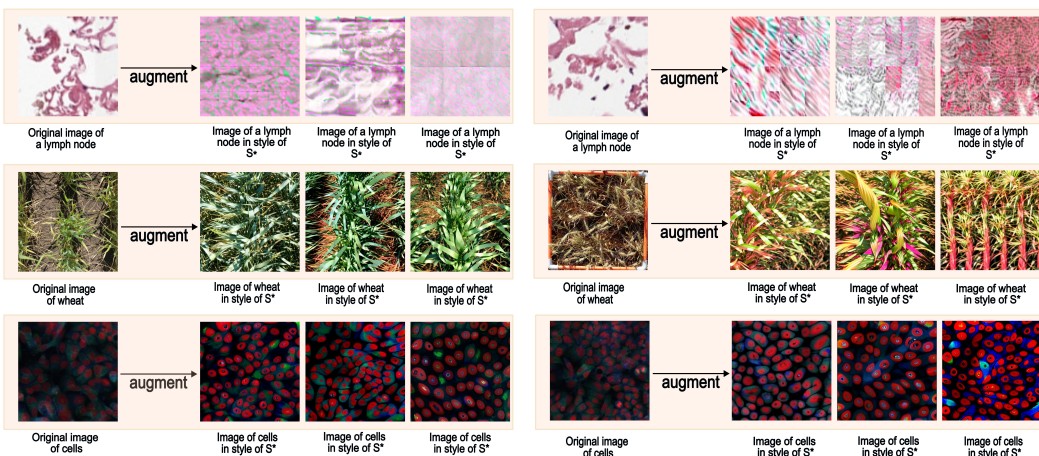

**Figure 14:** Text Inversion manipulation results for dramatically out-of-distribution data to Stable Diffusion training domain, as a domain adaptation approach. For each case, four sample images are randomly selected from the target test domain, and a style token $S_*$ is learnt with text inversion and used as a style prompt to augment the original training domain image. Images are manipulated with the Text Inversion prompt from the left first original image in each group of four images. The samples from top to bottom are 1) Histological image from Camelyon-17 (Bandi et al., 2018) 2) Cell image from RxRx1 (Taylor et al., 2019) 3) Wheat image from GlobalWheat (David et al., 2020; 2021).

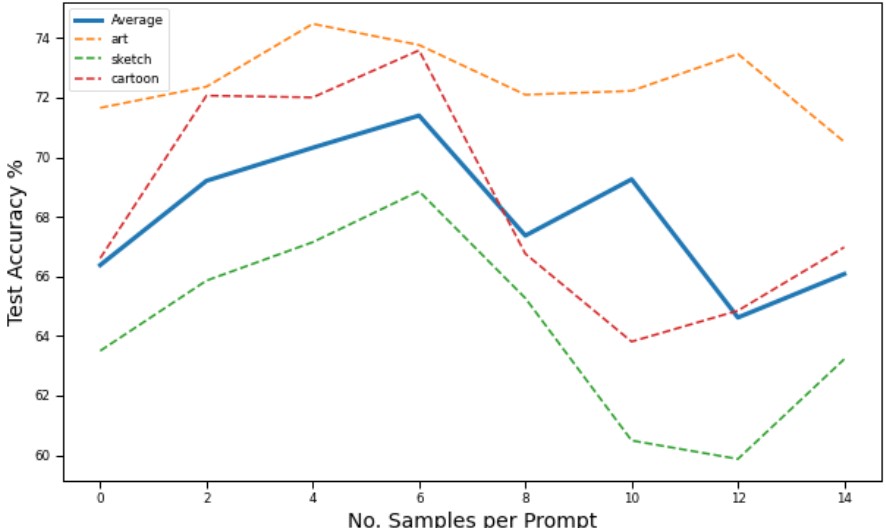

**Figure 15:** Number of samples generated for each prompt against test accuracy. The test accuracy is based on OURS(M/D) with ResNet-18 trained on Photo source domain.

similar to the query, so small differences in queries sometimes return highly variable results. For this reason, we try a variety of queries in an attempt to return images similar to the one that the generator produced to increase our confidence about the absence of a given image. While the first two examples (*"A corgi with a hat under water" and "A blue peacock cooking bacon in the kitchen"*) might be unlikely to occur in daily life, they might still occur in the context of captioning creative artworks (e.g., captioning of frames of animation movies or collage) and be useful to alleviate the reliance on spurious features (e.g., by perturbing the background or location in which an object is found). The last two examples (*"A cup in the amazon forest" and "A clock in the rain"*) exemplify much more common observations from the real world, that we could not retrieve from the training set. We also show SD can meaningfully manipulate synthetic images that cannot be found in the training set (see Fig. 13).

### I.2 QUALITATIVE EXAMPLES OF THE AUGMENTED IMAGES

In Fig. 12 we show additional examples of editing produced by Stable Diffusion. As it can be observed, Stable Diffusion may unintentionally manipulate features associated to the class label, without changing it. For instance, the augmented variants House and the Dog pictures in Fig. 12 significantly change their structure (e.g., structure of the house or breed of the dog), while preserving some similarities. Notice, this behavior is actually required when translating from domains with insufficient class-relevant information (e.g., when translating from a pencil sketch to a photo or painting, generators must infer color information).

### I.3 THE MORE THE (SYNTHETIC) DATA, THE BETTER?

While in our framework the diversity of interventional samples is controlled by prompting strategy, a natural question is whether generating more samples can be beneficial. Therefore, for the PACS experiment, we ablate the amount of images we generate for each target domain (see Fig. 15) using Minimal prompts and Stable Diffusion. As it can be seen, increasing the amount of generated images up to 6 per-domain produces a 1.52% increase in the performance. Adding more data seems to degrade the performance. We leave to future work understanding whether this is due to the shift induced by the inevitable artifacts or low-quality images that might be produced when increasing the amount of generated samples or by the potentially low variety in the generated results.

### I.4 QUALITATIVE EXAMPLES OF FAILURES

In Fig. 14 we present three failure cases of Stable Diffusion. In the first row, we observe Stable Diffusion fails at manipulating histological input images from the Camelyon-17 (Bandi et al., 2018) and the RxRx1 (Taylor et al., 2019) datasets. Camelyon-17 images contain tumoral and non-tumoral tissue captured in different environments. Since the changes between domains are hard to describe through language, we use Text inversion in order to learn how to transform from the source to the target domains. As it can be seen, Stable Diffusion fails to produce realistic samples in this setting, probably because the input images and text are well out-of-domain. A less severe failure occurs on RxRx1 (second row), which represents HUVEC cells. In this case, the generated images still result in a distortion of the input that makes them unrealistic. For the GlobalWheat (David et al., 2020; 2021) dataset, it is apparent that while Stable Diffusion can generate plants but it does not reproduce the specific species depicted in the original input and sometimes produces completely unrealistic instantiations of plants. This failure is particularly bad considering its training set contains several images of wheat crops; however, in those images, the crops are not captured from the angle in which they are captured in GlobalWheat (thus inducing a distribution shift). These failures suggest future research should be directed towards improving the ability of T2I generators to manipulate only the environmental variables for out-of-domain data, under the assumption a few text and image pairs from these unknown domains can be leveraged.

### I.5 THE DOMAIN SHIFT BETWEEN TARGET DOMAIN AND SYNTHETIC TARGET DOMAIN

Sometimes the target domain description cannot fully represent the domain features as prompts to the generative model. For example, we observe the "Sketch" domain of the PACS dataset and the synthetic "Sketch" Domain is visually different as shown in Fig. 16. This is mainly due to the bias in specific dataset collection processes and also the bias in the training data of the generative model, which introduce the discrepancy in understanding of some natural language concepts.

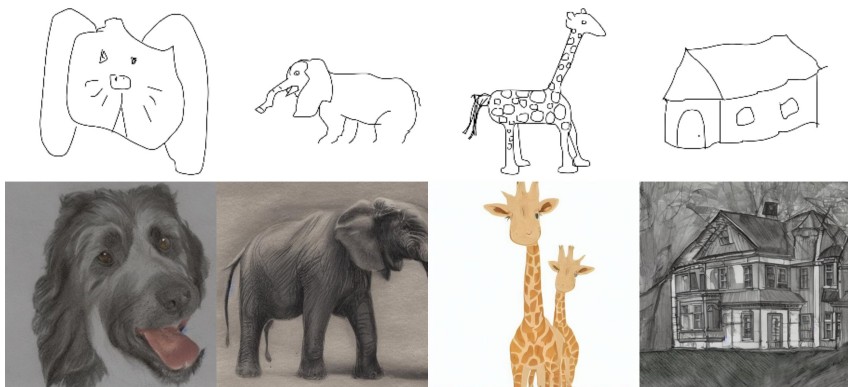

**Figure 16:** Comparison between *"Sketch"* domain in PACS and Stable Diffusion Synthetic Data. **Top:** Sample sketch images from PACS dataset. **Bottom:** Sample synthetic data generated with SDEdit.

## J IMAGE-GENERATION PROMPTS

We list the actual prompts used in all settings. The language enhancement prompts can either be generated by users following hint and language model specified in Sec. 3.2, or see our repo under *prompt* directory.

### J.1 PACS

PACS: prompt is set in format "[TEMPLATE] of [CLASS LABEL]". The templates are as follows:

1. Minimal: {'art painting':['an art painting of'],'sketch':['a sketch of'],'cartoon':['a cartoon of'],'photo':['a photo of']}

2. Hand-crafted: 'art painting': ['an oil painting of', 'a painting of', 'a fresco of', 'a colourful painting of', 'an abstract painting of', 'a naturalistic painting of', 'a stylised painting of', 'a watercolor painting of', 'an impressionist painting of', 'a cubist painting of', 'an expressionist painting

of ',' an artistic painting of '], 'sketch':['an ink pen sketch of ', 'a charcoal sketch of ', 'a black and white sketch', 'a pencil sketch of ', 'a rough sketch of ', 'a kid sketch of ', 'a notebook sketch of ','a simple quick sketch of '], 'photo': ['a photo of ', 'a picture of ', 'a polaroid photo of ', 'a black and white photo of ', 'a colourful photo of ', 'a realistic photo of '], 'cartoon': ['an anime drawing of ', 'a cartoon of ', 'a colorful cartoon of ', 'a black and white cartoon of ', 'a simple cartoon of ', 'a disney cartoon of ', 'a kid cartoon style of ']

3. Language Enhancement Moderate/Conservative: Generate with hint and language model specified in Sec. 3.2

## J.2 OFFICEHOME

1. Minimal: 'Art':['an art image of '],'Clipart':['a clipart image of '],'Product':['an product image of '],'Real World':['a real world image of ']
2. Handcrafted: 'Art':['a sketch of ', 'a painting of ', 'an artistic image of '],'Clipart':['a clipart image of '],'Product':['an image without background of '],'Real World':['a realistic photo of ']
3. Language Enhancement Moderate/Conservative: Generate with hint and language model specified in Sec. 3.2

## J.3 NICO++

1. Minimal: {'autumn':['autumn'],'dim':['dim'],'grass':['grass'],'outdoor':['outdoor'],'rock':['rock'], 'water': ['water']}
2. Hand-crafted: 'autumn': ['in autumn', 'autumn', 'autumn with fallen leaves'], 'dim':['during sunset','in the evening','twilight'], 'grass': ['on grass','on grass meadow', 'with grass'], 'outdoor': ['in outdoor environment','outdoor', 'in wild environment'], 'rock':['on the rock','rock','with rock'], 'water':['in water','under water','water']
3. Language Enhancement Moderate/Conservative: Generate with hint and language model specified in Sec. 3.2

## J.4 DOMAINNET

1.Minimal: 'real': ['a photo of '], 'clipart': ['a clipart of '], 'sketch': ['a sketch of '], 'infograph': ['a infograph of '], 'quickdraw': ['a quickdraw of '], 'painting': ['a painting of ']
2.Hand-crafted = 'real': ['a photo of ', 'realistic photo of '], 'clipart': ['a clipart of ', 'a prodcut image of '], 'sketch': ['a sketch of '], 'infograph': ['a infograph of '], 'quickdraw': ['a quickdraw of '], 'painting': ['a painting of ']
3. Language Enhancement Moderate/Conservative: Generate with hint and language model specified in Sec. 3.2

## J.5 IMAGENET-9

1. Hand-crafted:background:[" in a parking lot", " on a sidewalk", " on a tree root", " on the branch of a tree", " in an aquarium", " in front of a reef", " on the grass", " on a sofa", " in the sky", " in front of a cloud", " in a forest", " on a rock", " in front of a red-brick wall", " in a living room", " in a school class", " in a garden", " on the street", " in a river", " in a wetland", " held by a person", " on the top of a mountain", " in a nest", " in the desert", " on a meadow", " on the beach", " in the ocean", " in a plastic container", " in a box", " at a restaurant", " on a house roof", " in front of a chair", " on the floor", " in the lake", " in the woods", " in a snowy landscape", " in a rain puddle", " on a table", " in front of a window", " in a store", " in a blurred backround"]

## J.6 CELEBA-SUB

1. Hand-crafted experiment:
"blonde":["male"],"non-blonde":["female"]

## J.7 TEXTURE

We apply human-in-the-loop to iteratively improve the quality of prompt and augmentation in Texture dataset. We start with a set of heuristic prompt as original version. Then based on the image

generated, we add more representative prompts to further diversity the texture features. As shown in Tab. 16, by iteratively improving prompts, we achieve a final **8.28%** improvement more than **5.44%** of the initial improvement with respect to ERM.

1. Hand-crafted Final Version:

texture:['pointillism','rubin statue', 'rusty statue','ceramic','vaporwave','stained glass','wood statue','metal statue','bronze statue','iron statue','marble statue','stone statue','mosaic','furry','corel draw','simple sketch','stroke drawing', 'black ink painting','silhouette painting','black pen sketch','quickdraw sketch','grainy','surreal art','oil painting','fresco', 'naturalistic painting', 'stylised painting', 'watercolor painting', 'impressionist painting', 'cubist painting', 'expressionist painting','artistic painting']

2. Hand-crafted Original Version:

texture:['corel draw','simple sketch','stroke drawing', 'black ink painting','silhouette painting','black pen sketch','quickdraw sketch','grainy','surreal art','oil painting','fresco', 'naturalistic painting', 'stylised painting', 'watercolor painting', 'impressionist painting', 'cubist painting', 'expressionist painting','artistic painting']

