# OpenReview forum: "Not Just Pretty Pictures: Toward Interventional Data Augmentation Using Text-to-Image Generators"
_ICLR.cc/2024/Conference — Submitted to ICLR 2024_

### Official Review · Reviewer_3cd6 · 2023-10-30

**Soundness:** 3 good
**Presentation:** 3 good
**Contribution:** 2 fair
**Rating:** 5
**Confidence:** 4

**Summary:**

The proposes to use text-to-image generative models like Stable Diffusion for interventional data augmentation to simulate interventions over the environmental factors that are likely to change across domains. The authors argue that this interventional data augmentation would improve the generalization behavior of models over out-of-distribution data and reduce the reliance on spurious features during training. The two metrics that the authors measure are Single-Domain Generalization (SDG), which tests the generalization behavior to a new domain for instance natural image to sketch when trained only on the natural image domain, and Reducing Reliance on Spurious Features (RRSF) which measures the reliance on spurious features for model training such as relying on background to classify foreground.

For SDG, the authors use SDEdit on top of text-to-image models to generate source images in the target domain and use these generated images also for training. Interestingly the authors also show that instead of generating images in the specific target domain, similar performance can also be achieved if images are generated in a different set of target domains.

For RRSF again, the authors explicitly design prompts that try to reduce the effect of specific spurious correlations. For instance, generate images in various backgrounds to reduce the bias towards the background.

Across different datasets the authors show improved performance compared to various baseline approaches.

**Strengths:**

1. I like the idea of using SDEdit on top of text-to-image generative models to remove the model's biasness to spurious features and also generalizing to new domains.

2. The ablation study showing that a similar performance can be achieved for SDG even if images are are not generated for a specific target domain is nice and very useful.

**Weaknesses:**

1. In figure 3, it seems that Text2Image variant has the least biases across all three datasets. But the text on page 8 the authors suggest that "Text2Image seems to be less effective than other techniques in reducing background and texture biases". Can the authors clarify this?

2. The results in Figure 2 where the Text2Image variant performs better/comparable to SDEdit variant undermine the idea proposed in the paper.  In the Text2Image variant, there is no image conditioning for generating images. Several other papers[1] have also shown that using synthetic data from generative models improves the generalization performance of the classifiers. Where is the novelty then coming for this paper?

3. To further show that the generalization improves for unseen target domains, can the authors also show results for Cifar10-C and ImageNet-C datasets?

[1] Synthetic Data from Diffusion Models Improves ImageNet Classification. Azizi et al. https://arxiv.org/abs/2304.08466

**Questions:**

I have already mentioned my questions in the weakness section

---

> ### Author Response · Authors · 2023-11-17
>
> We thank Reviewer 3cd6 for their detailed summary and feedback. In particular, we appreciate their efforts in finding an important typo in our analysis, which we have now corrected (see Q1). We address their concerns below.
>
>
> **Q1: Figure 3 shows Text2Image is the best at RRSF, but statements on page 8 seem to contradict this.**
>
>
> We thank the reviewer for pointing out the typo. We fixed it in the revised draft (in red). Text2Image is indeed the most effective approach.
>
>
> **Q2.1: The effectiveness of Text2Image “undermine[s] the idea proposed in the paper”**
>
>
> Please refer to **General Response (B)**.
>
>
> **Q2.2: “Where is the novelty” relative to [1]?**
>
>
> The goal of [1] is to show:
> *  it is possible to fine-tune stable diffusion to produce a class-conditional generative model whose synthetic samples mimic the training distribution (goal: reduce gap between synthetic and real)
> * they test the performance on i.i.d samples with respect to the training set
>
> Our goal is to:
> * provide an extensive analysis of how to best use zero-shot generative techniques to simulate non-i.i.d. test domains (with no form of fine-tuning!)
> * we test the performance on non-i.i.d. test domains, showing zero-shot generated samples can be used to reduce bias and improve generalisation across a wide variety of domains
> * draw conclusions about how to most effectively exploit zero-shot generative capabilities of SD
> Thus, while [1] is an interesting related work (which is why we cite it in both Sections 1 and 2), the authors’ purpose, experimental setting, and results are all distinct from ours. For additional context, please consult our list of contributions in **General Response (C).**
>
>
>
>
> **Q3: “To further show that the generalization improves for unseen target domains, can the authors also show results for Cifar10-C and ImageNet-C datasets?”**
>
>
> We would like to point out that CIFAR-10-C and ImageNet-C are synthetic datasets obtained by applying some well-known parametric and handcrafted transformations to the images. Several of these transformations try to approximate the observation of different weather conditions or lighting conditions. We have experimented on 7 benchmarks, among which we have  two large scale dataset (NICO++ and DomainNet), which represent some of these transformations and are made of naturalistic data – real photos captured in, e.g., dim light, autumn weather, etc.
>
>
> However, as requested, we have also performed a comparable SDEdit experiment (using handcrafted prompts) on CIFAR-10-C:
>
>
> **Cifar-10-C Result with ResNet-18**
> | | Blurring Avg. | Noise Avg. | Compression Avg. | Transformations Avg. | Overall Avg. |
> |-----------|---------------|------------|------------------|---------------------|--------------|
> | ERM       | 72.04        | 74.01    | 75.71            | 73.93              | 73.55       |
> | MixUp  | 73.79 | 76.22 | 77.46 | 75.72 | 75.48 |
> | PixMix | 75.84 | 79.41 | 81.39 | 79.32 | 78.63 |
> | SDEdit      | 74.28        | 75.71    | 77.10            | 75.02              | 75.11       |
>
>
> It is important to note that Stable Diffusion (SD) is trained on web-scraped data, and as such we do not expect it to perform these kinds of artificial transformations as well as the naturalistic ones found in, e.g., NICO++. All the same, it appears that SD is still able to synthesize augmented data that can improve performance even on these unseen transformations. However, we note that, as these parametric transformations are artificially produced and mathematically well-understood (with implementations available in open-source libraries), it is obviously more efficient to target them with parametric augmentations at training time than with SD.

---

> > ### Comment · Reviewer_3cd6 · 2023-11-22
> > **Thanks for the response**
> >
> > While I thank the authors for conducting experients over ImageNet-C and Cifar10-C datasets, I think the fact that the "Text2Image" baseline performs better/comparable to SDEdit variant undermines the novelty of the paper. While the authors mention in their rebuttal, that the experimental setting and results are different from [1], I feel the main idea is still the same. Thus I am unwilling to increase my ratings.

---

> ### Author Response · Authors · 2023-11-22
> **Thanks for your response, an important clarification**
>
> We appreciate the reviewer taking time to respond to our rebuttal.
>
> **P1**: **the fact that the "Text2Image" baseline performs better/comparable to SDEdit variant undermines the novelty of the paper**
>
> We respectfully disagree, because our novelty is *not* to establish SDEdit as the best performing method, it is to study a range of image synthesis and editing techniques (including both SDEdit and Text2Image) in the context of SDG and RRSF. We make a more detailed argument on this point in **General Response (B)** (Conditioning and Interventions); but in sum: it is a contribution, not a contradiction, that our investigation explores both ways to simulate interventions and finds Text2Image to often outperform SDEdit.
>
> **P2.1: [with respect to] [1], I feel the main idea is still the same**
>
> The main similarity between [1] and our work is the use of foundational generative models to generate synthetic data to train classifiers. (We do not claim novelty on this point, and in Section 2, we provide a detailed discussion of additional works that are also similar to our work in this respect.) Below, we note several key differences between our work and [1].
>
> In [1], the authors show:
> - Assuming **access to all images of ImageNet**, Imagen can produce the best **approximation of ImageNet if fine-tuned** on it (according to FID, IS and CAS scores)
> - Adding the so-produced approximation of ImageNet to the ImageNet training set improves classification performance **on ImageNet itself**.
> - Although the authors talk of performing augmentations in the abstract, the generated data is explicitly generated to resemble ImageNet. **No deliberate manipulation of the image features is performed. The test set is i.i.d. with respect to the dataset**.
>
> In our work:
> - Assuming **no access to non-i.i.d. test domains training data**, Stable Diffusion can approximate them **zero-shot** if some **meta-data** is accessible. No image is accessed from the non-i.i.d. test domains. This is a significantly more difficult problem that **cannot be solved with more available i.i.d. training data (like [1] does)**. If, similarly to [1], we assumed full access to the test domain (non-i.i.d., in our case), the problem would not even require a generative model to be solved with perfect accuracy on PACS/Office-Home.
> - Adding the zero-shot generated non-i.i.d. data to i.i.d. data, we improve the performance **on non-i.i.d. test data** (not accessible for any form of fine-tuning) and reduce the impact of **biases** on model's predictions.
> - Our experiments in the main paper and rebuttal (Table 2 in the paper,  Q2 response to reviewer MWyF) show **exact knowledge of the meta-data is not required**.
> - We ablate and analyse which conditioning, prompting and filtering techniques produce the best performance, and compare with a retrieval baseline pointing out weaknesses and strengths of all approaches in different settings, as already extensively discussed in **General Response (C)** (Novelty and Contribution).
>
> **P2.2: the experimental setting and results are different to [1], I feel the main idea is still the same"**
>
> The different experimental setting is not a minor difference. The availability of i.i.d. training data and testing only on i.i.d. training data vs. the non-availability of non-i.i.d. data and testing on non-i.i.d. test data is the fundamental difference that distinguishes in-domain generalization (which [1] addresses) and out-of-domain generalization and reducing the reliance on spurious features (which we address). The two fields are fundamentally different areas of research: solutions and conclusions that hold in one do not necessarily generalize to the others. Indeed, *under the assumptions of our setting, [1] would not be applicable (as there would be no non-i.i.d. data to fine-tune on).*
>
> Having carefully read [1], it is unclear why the novelty of the results of our work are diminished by paper [1], as they operate in very different settings with different methodologies, goals, and contributions.

---

### Official Review · Reviewer_nrcC · 2023-10-30

**Soundness:** 2 fair
**Presentation:** 3 good
**Contribution:** 3 good
**Rating:** 6
**Confidence:** 5

**Summary:**

This article carried out the first investigation of T2I generators as general-purpose interventionaldata augmentation mechanisms, showing their potential across diverse target domains and potential downstream applications.
• Authors perform extensive analyses over key dimensions of T2I generation, finding the conditioning mechanism to be the most important factor in IDA.
• Authors show that interventional prompts are also important to IDA performance; but in in contrast with previous works, we find that post-hoc filtering is not consistently beneficial.

Generally, this article describe why text to image generator from stable diffusion outperforms others methods.

**Strengths:**

This article carried out the first investigation of T2I generators as general-purpose interventionaldata augmentation mechanisms, showing their potential across diverse target domains and potential downstream applications.
• Authors perform extensive analyses over key dimensions of T2I generation, finding the conditioning mechanism to be the most important factor in IDA.
• Authors show that interventional prompts are also important to IDA performance; but in in contrast with previous works, we find that post-hoc filtering is not consistently beneficial.

This work generally makes efforts on data shift problem. A good mind in solving data augmentation problem.

**Weaknesses:**

Not very soundness from technical side. No novelty in the model is presented.

**Questions:**

Have you ever compared results with some other generative methods like GAN?

---

> ### Author Response · Authors · 2023-11-17
>
> We thank Reviewer nrcC for acknowledging the clear strengths and contribution of our work. We respond to their remaining concerns below.
>
>
> **Q1.1: “Not very soundness from technical side.”**
>
>
> Our experiments use several well-known and widely-utilized methodologies (e.g., Stable Diffusion-based image generation techniques including Text2Image, SDEdit, ControlNet, etc.). We are confident in the soundness of our implementation, which we have anonymized and made available at https://tinyurl.com/49sb675n. We are happy to engage in further discussion or experimentation to address any specific technical concerns the reviewer might have.
>
>
> **Q1.2: “No novelty in the model is presented.”**
>
>
> Our goal in this work is not to develop a novel methodology or model for performing interventional data augmentation (IDA). Rather, it is to:
> 1. Analyse the extent to which generative diffusion models, when used in a zero-shot setting, can serve as a general-purpose IDA mechanism.
> 2. For each application case (SDG, RRSF), discern the factors that contribute to their superior or inferior effectiveness (conditioning, filtering, prompting).
> No previous work has conducted the necessary study to address these questions, and our extensive findings, as described in **General Response (C)**, are indeed novel.
>
>
> **Q2: Have you compared with GANs?**
>
>
> We thank the reviewer for this suggestion. In response, we carry out our SDG PACS experiment using VQGAN-CLIP [1] as a comparison and include the result in the selected table below and **Table 3**. We observe it underperforms with respect to Stable Diffusion. While this should not discourage future work from considering newly developed GANs, this reflects the superior generative capabilities of diffusion models established in the current literature [2].
>
>
> **SDG PACS result with ResNet-18 for VQGAN-CLIP comparison**
>
>
> |              | Art    | Photo  | Sketch | Cartoon | Average |
> |--------------|--------|--------|--------|---------|---------|
> | ERM          | 74.8   | 39.67  | 48.12  | 72.37   | 58.74   |
> | VQGAN-CLIP(M)| 78.09  | 54.38  | 53.78  | 77.76   | 66.00   |
> | SDEdit(M)    | 82.27  | 58.87  | 72.76  | 81.93   | 73.96   |
>
>
>
>
> [1] Crowson, Katherine, et al. "Vqgan-clip: Open domain image generation and editing with natural language guidance." *European Conference on Computer Vision.* Cham: Springer Nature Switzerland, 2022.
>
> [2] Rombach, Robin, et al. "High-resolution image synthesis with latent diffusion models." *Proceedings of the IEEE/CVF conference on computer vision and pattern recognition.* 2022.

---

> > ### Author Response · Authors · 2023-11-22
> > **About the discussion phase**
> >
> > We thank the reviewer for their work in examining our paper.
> >
> > We hope our additional experiment on GANs satisfies the reviewer's question and our explanation about the scope of our work is clear.
> >
> > We would appreciate it if the reviewer could have a look at our rebuttal and let us know if their concerns have been addressed before the conclusion of the discussion phase, so that we can address any other concerns.
> >
> > Best Regards,
> > The Authors

---

### Official Review · Reviewer_MWyF · 2023-10-31

**Soundness:** 3 good
**Presentation:** 3 good
**Contribution:** 2 fair
**Rating:** 5
**Confidence:** 4

**Summary:**

The paper uses text-to-image generators and editing techniques to generate training data. Experiments were performed for domain generalization benchmarks with supportive results. There were extensive ablations and analysis over types of prompts, conditioning mechanisms, post-hoc filtering and editing techniques.

**Strengths:**

- The paper has extensive experiments and ablations.
- The analysis comparing different editing methods is insightful.

**Weaknesses:**

1. Generalizability of the method
    - Almost all the results seem to assume that the target domain can be easily described and that the number of domains are known. However, this does not always hold. E.g., in iwildcam, where the target domain consists of images from different camera traps resulting in different locations, viewpoints, etc., it may not be obvious how to describe the target domain.
    - Furthermore, the proposed method on “Breaking spurious correlations” (Fig 3) requires a human to hand craft prompts which may be expensive to attain.
    - E.g., [1] uses a captioning model to describe the data, then gpt to summarize into domain descriptions. These descriptions are then used in the prompts. Thus, it doesnt require knowledge of the domains.
2. "Describing the Target Domain Is Not Necessary”. Table 2.
    - It is not clear to me what the message of table 2 is. As it is without target domain information, it seems say that SD is biased towards generating certain domains and those domains happen to be aligned with the target for this dataset, but this may not be the case for other datasets.
3. From A.1, it seems like there was different number of additional data for generated images and baseline augmentation techniques. Can the authors explain this choice? It may be interesting to see how the performance changes with amount of data similar to (He et al., 2023;  Sariyildiz et al., 2023).
4. The conclusions of the paper seems similar to that of (Bansal & Grover, 2023) who also used pre-defined text prompts to generate data. They also showed that a combination of real and generated data results in better performance, although on IN-Sketch and IN-R. The evaluation setup may be slightly different but the conclusions from SDG seems to be similar.



[1] Diversify Your Vision Datasets with Automatic Diffusion-based Augmentation. NeurIPS’23

**Questions:**

Other than the questions raised above:
- What is the performance of Retrieval on DomainNet in Fig 5?
- I would suggest moving some technical details, e.g. the setup, how many images are generated for each original image, a brief description of how is retrieval done, to the main paper.
- It may be useful to have an additional column in the table of results for the runtimes. The baseline augmentations should be much cheaper to attain than generating data with SD.

---

> ### Author Response · Authors · 2023-11-17
>
> We thank Reviewer MWyF for their detailed feedback. The potential weaknesses they outline are important to address, and we believe that our responses below are sufficient to address each concern.
>
>
> **Q1.1a: Assumption that “the target domains can be easily described.”**
>
>
> We agree that, in principle, not all domain shifts can be easily described via language; however our work clearly targets those that can. These represent a wide variety of problems of interest (we use 7 of the most popular benchmarks in the domain-shift literature). Indeed, reference [1] suggested by the reviewer provides further evidence that even when they are not easily describable, language-driven generation can achieve impressive performance on 3 additional benchmarks. In contrast, none of the non-textual baselines are able to outperform ERM on more than a small subset of the 7 benchmarks considered by our work, clearly indicating they are less general.
>
>
> **Q1.1b: Assumption that “the number of domains [is] known.”**
>
>
> While it is true that our primary SGD experiments assume that the number of domains is known, we have several key experiments (experiments in Table 2, the additional NICO++ we include in response to Q2 for SGD, and all RRSF experiments) that do not.
>
>
> **Q1.2: Human handcrafted prompts required? Expensive to attain?**
>
>
> For SDG, we experiment extensively with two prompting strategies that do not require human input: language enhancement (LE) and minimal (M). The same is very likely possible for RRSF, but for this task, we elected to focus on ablating the conditioning mechanism instead of the prompting strategy, as our results from SDG showed this to be more important. Regarding expense, we note that the handcrafted prompts took only a single participant 2-5 minutes per benchmark.  (across all 3 RRSF benchmarks).
>
>
> **Q1.3: [1] does not “require knowledge of the domains.”**
>
>
> We thank the reviewer for mentioning [1]. However, we kindly note that : [1] assumes access to images of several domains to create the captions. Indeed, this is much more information than we access for, e.g., Minimal (M) prompts, which are produced with meta-data only. However, this requires explicit access to the target domain images, which is unavailable in our domain generalization setting. We will discuss the work in the revised draft.
>
>
> [1] Diversify Your Vision Datasets with Automatic Diffusion-based Augmentation. NeurIPS’23
>
>
> **Q2: Table 2 seems to say that SD is biased to generate some domains that happen to be aligned with the target for this dataset, but may not be the case for other datasets.**
>
>
> We perform the experiment of Table 2 for NICO++, which has domains that are significantly not aligned (see the column names in the Table below). This clearly indicates the intervention performed aids generalization per-se as the improvement is substantially maintained when the target domain is not included.
>
>
> **SDG NICO++ Result with ResNet-50 for comparing the effect of accessing synthetic target domain or not**
>
>
> |              | autumn | dim    | grass  | outdoor| rock   | water  | Average |
> |--------------|--------|--------|--------|--------|--------|--------|---------|
> | ERM          | 66.74  | 70.37  | 72.05  | 71.3   | 66.58  | 72.64  | 69.95   |
> | SDEdit(M) ×  | 67.90  | 71.70  | 72.61  | 72.32  | 67.10  | 73.79  | 70.90   |
> | SDEdit(M) ✓  | 68.28  | 71.42  | 72.68  | 72.31  | 67.95  | 74.07  | 71.12   |
>
>
> **Q3.1: Why “different number of additional data for generated images and baseline[s]”?**
>
>
> The difference is caused by the current cost of generating samples with Stable Diffusion (SD). Since the process is expensive, we perform it offline before training:
> * the augmentation baselines produce $D*E$ augmentations where $E= 50$ (the number of epochs), and $D$ is the size of the training set.
> * the SD-based approaches generate $D*(C-1)$ samples, where $C$ is the number of domains. $C=4$ or $C=6$ across the SDG experiments.
> * Since $(C-1)$ is much lower than 50, SD produces much less augmentations
> The superior performance of using fewer SD-based augmentations indicates interventions targeting the right environmental variables are significantly more effective than generic   augmentations whose contribution to generalization is unclear.
>
>
> **Q3.2: How much does performance change with amounts of data similar to cited works?**
>
>
> In **Appendix I.3** we already report this result for PACS. The conclusion is that a few additional data samples can help, but too much data can degrade the performance. The fact that few additional samples are enough to outperform all SOTA baselines indicates the slow generation is counterbalanced by the effectiveness of using just a few generated samples.

---

> > ### Author Response · Authors · 2023-11-17
> >
> > **Q4: Evaluation setup different but similar conclusions to Bansal & Grover (2023)?**
> >
> > Bensal and Grover (2023) do not try to produce deliberate manipulations of the environmental variables, as we do. They only apply CLIP prompts to generate the data, which, only as a side-effect, results to be robust to some forms of distribution shift.
> > Their contributions do not overlap with any of our contributions listed in **General Response (C).**
> >
> > **Q6: What is the performance of Retrieval on DomainNet in Fig 5?**
> >
> >
> > Thank you for drawing attention to this issue: the lack of Retrieval in DomainNet was a plotting mistake, which we have corrected in the updated draft.
> >
> >
> > **Q7: “[S]uggest moving some technical details, e.g. the setup, how many images are generated for each original image, a brief description of how is retrieval done, to the main paper.”**
> >
> >
> >  Thank you for your suggestion. We agree and plan to update the manuscript accordingly in the camera-ready draft.
> >
> >
> > **Q8: It may be useful to have an additional column in the table of results for the runtimes. The baseline augmentations should be much cheaper to attain than generating data with SD.**
> >
> >
> > We report the runtime for generation in **Appendix G**. As we already discussed the generation time is a downside, hence we perform it offline once and can be reused across models and experiments. If performed offline, loading the generated from disk is significantly faster than competing methods. We also observe:
> > * Reducing SD generation time is an active area of research. For instance, passing from version 1.4 and upgrading to PyTorch 2.0 and version 1.5 of stable diffusion, with adequate batching, we have already observed a 150% reduction in generation time. Making the process more efficient is out of the scope of this paper. Furthermore, the offline generative process can be easily parallelised.
> > * Given the flexibility and effectiveness of the procedure, paying the price for generation can be deemed convenient if compute is available and before better and faster techniques are proposed.

---

> > > ### Author Response · Authors · 2023-11-22
> > > **About the discussion phase**
> > >
> > > We thank the reviewer for their work in examining our paper.
> > >
> > > We hope our additional experiment on NICO++, the discussion of related works satisfies the reviewer's question and our explanation about the scope and additional details of our work are clear.
> > >
> > > We would appreciate it if the reviewer could have a look at our rebuttal and let us know if their concerns have been addressed before the conclusion of the discussion phase, so that we can address any other concerns.
> > >
> > > Best Regards,
> > > The Authors

---

> ### Comment · Reviewer_MWyF · 2023-11-22
>
> Thanks for the clarifications and additional experiments!
>
> About Table 2, my point was more general, if the target domain were from another dataset or with substantial shifts, it is unlikely that the transformed training data would help. Thus, it seems to be with some assumptions that "Describing the Target Domain Is Not Necessary".
>
> I have read the rebuttal and I appreciate the discussions. However, I am inclined to keep my score.

---

> > ### Author Response · Authors · 2023-11-22
> >
> > Dear Reviewer MWyF,
> >
> > Thank you for your response. We will keep polishing our manuscript based on your suggestion.
> >
> > Best Regards,
> >
> > All Authors

---

### Official Review · Reviewer_bsQg · 2023-10-31

**Soundness:** 2 fair
**Presentation:** 3 good
**Contribution:** 2 fair
**Rating:** 6
**Confidence:** 4

**Summary:**

Utility of Text-to-Image generators for interventional data augmentation (IDA) toward improving single domain generalization (SDG) and reducing reliance on spurious features (RRSF) is the subject of this work.
Previous works studied using generators for generating training data; the contribution of this work is a deeper study of the same for SDG and RRSF tasks.

The work is a thorough study with many tasks and ablation. Although I believe the paper do not have any surprising finding and ranks low on novelty, it could be of interest to the research community.
However, I have many questions regarding their setup, which muddled their contributions quite a bit. My assessment therefore is a placeholder at the moment, and would likely change.

**Strengths:**

- Thorough study with four SDG and three RRSF tasks. Comprehensive evaluation with previous augmentation procedures and various image generators.
- Writing and presentation of results is easy to follow. I enjoyed comparisons made using simple baselines.

**Weaknesses:**

- Novelty of the paper is somewhat limited.

Please see questions.

**Questions:**

**Premise compromised?** The paper started with the premise that IDA is known to be useful for SDG and RRSF, and proceeded with two-fold objective of evaluating T2I generators and establishing a new state-of-art on SDG and RRSF.
However, as observed from Fig. 3 and 5, text2Image and retrieval baselines performed the best, which are both non-interventional augmentations. What then is the role of IDA and conditional generators?
Clearly stating the contributions can help. Is the paper suggesting to only evaluate unconditioned image generators using the task?

**Table 2** results are very interesting. Few questions.
1. The performance on the sketch domain is better without simulated target domain, why is that?
2. For comparison, could you please include the baselines: (a) ERM trained on all but target domain, (b) ERM trained on all the domains, (c) ERM trained only on the target domain.
3. It is intriguing that the performance is comparable even without simulated target domain for SDEdit, but none of the other target-agnostic augmentation are even close, why is that?
4. I suspect if there are any implementation differences between SDEdit and others (MixUp, CutMix etc.) causing the massive improvement (a common and annoying problem with PACS and other datasets), releasing your implementation can help. Also, can you add to the table (2) the performance of SDEdit with even more irrelevant prompts? How about if we use the prompts from OfficeHome on PACS dataset? How does the performance compare then?

**More information on prompts.** Could you please provide more information on the prompts used for generatring images on the three RRSF tasks? They are more nontrivial than for SDG, and yet their description is rushed in the main paper.
Overall, how much effort was spent on engineering the prompts and how were they tuned?

**Conclusion and contributions**. I am somewhat lost on the takeaways. Please spell them out. As I see it, conditioning of generators (since text2Image and retrieval work just as well) is not so important but the conclusion says otherwise.
What are the implications for evaluation of generators and SDG/RRSF research?

---

> ### Author Response · Authors · 2023-11-17
>
> We thank Reviewer bsQg for their comprehensive feedback, openness to improving their rating, and for clearly articulating their questions in detail. We believe the feedback helps us to further clarify the purpose of our work, and hope our response will stimulate an engaging and fruitful discussion.
>
>
> **Q1.1: “Premise Compromised?” Core premises and goals are unclear.**
>
>
> Please refer to **General Response (A).**
>
>
> **Q1.2 Is Text2Image interventional? Is it conditional?**
>
>
> Please refer to **General Response (B).**
>
>
> **Q1.3: Clearly state the contributions.**
>
>
> Please refer to **General Response (C).**
>
>
> **Q2.1 Why is sketch-domain performance “better without simulated target domain”?**
>
>
> The distribution of sketches produced by Stable Diffusion is substantially different from those represented in PACS (see **Appendix I.5, Figure 16** of our updated draft). Since tuning the prompts to align the Stable Diffusion and PACS distributions is out of the scope of our analysis (see our response to Q3 below), this behavior is not unexpected.
>
>
> **Q2.2 “[I]nclude the baselines: (a) ERM trained on all but target domain, (b) ERM trained on all the domains, (c) ERM trained only on the target domain.”**
>
>
> We perform the requested experiments and report the results in the Table below (and in our updated manuscript, **Table 14**). Before consulting the results, please note the following:
> 1. Training on the target domains (b,c) is nonstandard in the context of SDG, as they are i.i.d. to the test sets. Thus, (b,c) do not constitute a baseline but an empirical “upper bound” on the best possible interventional augmentation strategy (i.e., where the generator can directly sample from a perfect approximation of the target distribution).
> 2. Experiment (a) constitutes a typical Multi-Domain Generalization (MDG) setting that is not comparable to the SDG we focus on, as the reported metrics for the two kinds of experiments are different:
>    * In MDG, ERM is trained on all but the target domain and the performance on the unseen target domain is reported (column name is unseen target domain)
>    * In SDG, the average of the performance across 3 unseen target domains is reported for each known source domain (column name is the only known training domain).
> Therefore the MDG results cannot be compared with the ones in Table 2 of the paper, so for a fair comparison with (a), we use SDEdit in the context of MDG (i.e., SDEdit(MDG)) using minimal prompts.
>
>
> Our results are as follows:
> * Unsurprisingly, (b,c) achieve near-perfect performance, confirming that the train and test datasets are nearly perfectly i.i.d.
> * The performance improvement of SDEdit (MDG) with respect to (a) indicates the effectiveness of IDA also extends to the case where multiple training domains are available.
>
>
> |    | Art    | Photo  | Sketch | Cartoon | Average |
> |---------------|--------|--------|--------|---------|---------|
> | (b)  | 99.65  | 99.94  | 99.64  | 99.66   | 99.72   |
> | (c)    | 99.71  | 99.70  | 99.84  | 99.60   | 99.74   |
> | (a)    | 80.01  | 96.28  | 73.86  | 76.28   | 81.61   |
> | SDEdit (MDG)    | 87.5   | 95.75  | 79.21  | 85.2    | 86.91   |
>
> **Q2.3: Why is SDEdit so much better than target-agnostic augmentation[s]?**
>
> Our results indicate that the target-agnostic augmentations are:
> * Not as effective as commonly believed: most of them have been tested on benchmarks crafted from CIFAR and ImageNet but be less effective in other cases. On these datasets, a few of the methods we consider have already been shown not to be significantly more effective than ERM in the DomainBed benchmark (https://tinyurl.com/yeyjzvd9)
> * Not as “target-agnostic” as commonly believed: our RRSF experiments clearly show they explicitly target specific types of invariances while neglecting others.
> On the other hand, SDEdit can manipulate shape, texture and other factors in more sophisticated ways, as captured by the 4 SDG and 3 RRSF benchmarks we experiment with.
>
> **Q2.4.1 Concerns about implementation.**
>
> We have integrated all baselines in the same codebase and ensured no method has an unfair edge over the others, using the same evaluation code across all methods. We have anonymized our codebase and made it available for review at https://tinyurl.com/49sb675n, and will release the de-anonymized codebase upon acceptance.
>
> **Q2.4.2 Can you experiment with using OfficeHome prompts on PACS?**
>
> We appreciate the interesting suggestion for this experiment. We added the prompts for Officehome to the ones of PACS to augment the PACS dataset. We include the results in the table below (and in **Table 4** of the revised draft) indicating this method as SDEdit(PO-M). Note that adding these “irrelevant” prompts further improves performance.
>
> |            | Art   | Photo | Sketch | Cartoon | Average |
> |------------|-------|-------|--------|---------|---------|
> | SDEdit(PO-M)| 82.41 | 65.43 | 80.3   | 85.29   | 78.36   |
> | SDEdit(M)   | 82.67 | 62.94 | 73.78  | 86.33   | 76.43   |

---

> ### Author Response · Authors · 2023-11-17
>
> **Q3: “[P]lease provide more information on the prompts used for generating images on the three RRSF tasks [...] How much effort was spent on engineering them? How were they tuned?”**
>
> We asked a human participant to write up a set of prompts that would manipulate a specific feature of an image. We provided them only with information from metadata contained in the respective benchmark papers and the following high-level instructions:
> * For the background-bias prompts, we asked them to specify common image backgrounds randomly.
> * For the texture-bias prompts, we asked them to reason about what kind of transformations would change the texture of an object. Later, we asked the same participant to add a few additional prompts following the same instructions (see **Appendix F.1**).
> * For the demographic bias, using the meta-data of the task was enough to create “Minimal” prompts (i.e., considering all possible combinations of gender/color), so no participant was required.
>
> The prompts were not engineered nor tuned on any metric: our goal was to evaluate how well these systems perform “off-the-shelf” (i.e., where no time was spent engineering or tuning them). This illustrates the usefulness, flexibility, and simplicity of using off-the-shelf T2I generators for RRSF.
>
>
> **Q4: “Please spell out [the takeaways …] What are the implications for evaluation of generators and SDG/RRSF research?”**
>
>
> Please refer to our answers to **General Response (B)** (for context on our use of “conditioning” and “IDA”) and **General Response (C)** (for our response to this question).

---

> > ### Author Response · Authors · 2023-11-22
> > **About the discussion phase**
> >
> > We thank the reviewer for their work in examining our paper.
> >
> > We hope our additional experiments address the reviewer's questions and our explanation about the scope and additional details of our work are clear.
> >
> > We would appreciate it if the reviewer could have a look at our rebuttal and let us know if their concerns have been addressed before the conclusion of the discussion phase, so that we can address any other concerns.
> >
> > Best Regards,
> > The Authors

---

> > > ### Comment · Reviewer_bsQg · 2023-11-22
> > >
> > > Thanks for your response and patience. I am glad to find that the prompts used for RRSF task are not engineered. Also my implementation related concerns are answered well. Thanks for reporting teh performance of the requested baselines.
> > >
> > > Few concerns still remain.
> > >
> > > The abstract has the following sentence "A general interventional data augmentation (IDA) mechanism that simulates arbitrary interventions over spurious variables has often been conjectured as a theoretical solution to this problem and approximated to varying degrees of success". This statement is in contradiction to the 2nd contribution from the general response "Text2Image (conditioning only on text) and Retrieval generally outperform SDEdit (conditioning on both image and text), indicating that augmenting the original image is not necessary."
> > >
> > > The contribution lies in evaluating T2I models for SDG and RRSF. The takeaway from the paper is that we can suppress both the problems by simple methods like Retrieval or Text2Image (which is simple because it is only conditioned on the text). In that regard, the paper establishes a new state-of-the-art performance on many of the standard becnhmarks, which I appreciate. Especially since the new numbers are far greater than the previous best. But it seems like the improvement is coming from diverse datasets in the case of SDG (which is why target agnostic, i.e. SDEdit (M) $\times$ from Table 2 or using irrelevat OfficeHome prompts on PACs performed well). In other words, SDG with the datasets considered are not challenging enough because all the datasets are responding to diversified but label or task relevant data
> > >
> > > The experiments of the paper are sound, but the authors need to tone down unsupported claims made in the paper. SDEdit improvement over other target-agnostic baselines like MixUp is not because SDEdit made effective interventions instead it is likely because SDEdit generated novel images.  Similarly, the point about intervention as a solution for DG must be carefully addressed. Authors must also address the limitations of prompting (for instance, by inluding a failure case of SDG/RRSF) because it is not easy to descibe the the domain shift  that the T2I model also understands. Many of the distribution shift examples from the WILDS dataset are of that kind.
> > >
> > > For the above stated reasons, I cannot recommend an eager accept of the paper. Nevertheless, the paper did a good job of evaluating many simple baselines and in establishing state-of-the-art on many benchmarks. I am changing my score to 6.

---

> ### Author Response · Authors · 2023-11-22
> **Thank you for the response, about the remaining concerns.**
>
> Dear Reviewer bsQg,
>
> We sincerely thank you for raising your rating. Thank you for recognizing our **contribution in evaluating T2I models for SDG and RRSF and experiments and acknowledging the experiments are sound**. We have indeed experimented with large-scale and challenging datasets with a wide variety of classes like NICO++ (61K imgs, 60 classes) and DomainNet (410K imgs, 345 classes; one of the most challenging benchmarks in the domain due to its outstanding **complexity and scale**) to an extent that is rarely seen in RRSF and SDG literature.
>
> Regarding the **tension between the following two statements**:
> 1. “A general IDA mechanism […] has often been conjectured as a theoretical solution” (in the abstract)
> 2. “Text2Image (conditioning only on text) and Retrieval generally outperform SDEdit […] indicating that augmenting the original image is not necessary.” (in our General Response)
>
> Specifically, the tension is this: our empirical findings contrast with earlier theoretical literature, in that we generally find that “augmenting the original image is not necessary.” In other words, when we empirically test the strength of the conjectured theoretical solution (e.g., an approach like SDEdit), we find that there are alternatives (e.g., an approach like Text2Image) that perform even better without direct augmentation (i.e., not requiring sample-level alignment, per **General Response (B)**). This is a key contribution of our work: it unearths an empirical limitation of the theoretical literature, pointing to the need for additional theoretical analysis in the context of IDA.
>
> **We will make this key point of tension and the associated empirical contribution clearer in our abstract and introduction.** We will also strive for greater clarity on the remaining points you raised and will remove ambiguous statements.
>
> About the **limitations of prompting**, we had already included a failure case in Appendix (on 2 datasets from the WILDS dataset mentioned by the reviewer, please refer to Figure 14) but rest assured we will emphasise this aspect more in the main paper.
>
> Thank you again for your careful analysis of our response, we hope these lines **address the remaining concerns** about the framing of our work. Given the little time before the end of the discussion period, we will not be able to update the draft immediately to reflect these changes, but **all the points discussed above will be modified considering your suggestions in the final draft.**
>
> Best Regards,
>
> All Authors

---

### Author Response · Authors · 2023-11-17

We thank all reviewers for their helpful suggestions and insightful feedback. Given some concerns raised by more than one reviewer, in this General Response we **A)** clearly outline the premises and goals of our work, **B)** clarify the misunderstanding about what we mean by intervention, **C)** clarify our novelty and contributions.  We will use these points to further clarify the exposition of our paper. We will respond to each reviewer below and reference this response where relevant.

**A) Premises and Goals.**

For clarity, we enumerate our core premises and goals regarding interventional data augmentation (IDA). Our premises are that:

1. Substantial theoretical literature has hypothesized that general-purpose IDA could help solve SGD/RRSF (Ilse et al., 2021; Wang et al., 2022c; Wang & Veitch, 2022).

2. The usefulness of IDA has been demonstrated in limited empirical settings (Hong et al., 2021; Li et al., 2020; Ilse et al., 2021; Ouyang et al., 2022; Gowda et al., 2021).

3. There is empirical literature interested in using T2I generators such as Stable Diffusion to synthesize data to train more robust classifiers (Bansal & Grover, 2023; Azizi et al., 2023; He et al., 2023; Trabucco et al., 2023).

Our goals are to:
1. Understand the extent to which such models, when used in a zero-shot setting, can serve to simulate interventions.
2. For each application case (SDG, RRSF), discern the factors contributing to their superior or inferior effectiveness (e.g., conditioning, filtering, and prompting).

**B) Conditioning and Interventions.**

Text2Image, SDEdit, ControlNet, and InstructPix2Pix are all conditional generation methods:
* T2I conditions only on text
* SDEdit and InstructPix2Pix condition on both text and images
* ControlNet conditions on text, images, and “canny edge” maps

They are also all approximating an intervention on the environmental variables of a distribution, but at different levels. To illustrate this concept, consider the following example about whether smoking causes cancer:

* Case 1 (Sample-level alignment): clone each participant twice, making one clone smoke and one abstain, and measure the likelihood of each group developing cancer. This is clearly infeasible (and unethical), but there is another way to study whether smoking causes cancer (Case 2).
* Case 2 (Distribution-level alignment): given a population of smokers and non-smokers, measure their likelihood of developing cancer (while controlling for other population-level statistics that might be related). This is still interventional data collection, even though there is no sample-level alignment.

We investigate how to simulate the collection of interventional data using an image synthesis process.

* SDEdit is analogous to Case 1: we approximate image interventions over the experimental variable (e.g., domain).
* Text2Image is analogous to Case 2: we approximate sampling from different (interventional) distributions, but individual samples are not required to be aligned.

Given these observations and the previously stated goals, we find:

* SDEdit (cf. Case 1) outperforms all previous state-of-the-art augmentation baselines, indicating it as an effective IDA mechanism.
* Text2Image (cf. Case 2) is often superior to SDEdit (presumably due to the imperfection of the editing procedures) although it does not augment individual samples.

It is a contribution, not a contradiction, that our investigation explores both ways to simulate interventions and finds Text2Image to often outperform SDEdit. We will clarify this further in the final revision of the draft.


**C) Novelty and Contribution.**

Given our previous responses (A, B), we spell out several key contributions more clearly below. We find that:
* T2I-based editing techniques (e.g., SDEdit) outperform all the previously existing state-of-the-art augmentation techniques across 7 benchmarks differing in complexity and variety of environmental conditions.
* Text2Image (conditioning only on text) and Retrieval generally outperform SDEdit (conditioning on both image and text), indicating that augmenting the original image is not necessary.
* The choice of the conditioning technique can significantly affect performance.
* The choice of prompting, although important, is not fundamental. There’s no need to carefully tune the prompts.
* Given the strength of current generators, filtering out low-quality samples is not necessary (in contrast to previous literature using earlier generators).
* Between Text2Image and Retrieval, no method dominates the other in all cases. This suggests some evaluation tasks may be more aligned with Retrieval while others may benefit more from the generative capabilities of models. For instance, a benchmark with extremely unusual combinations of objects may benefit from generative capabilities and vice versa.

These findings are novel: no previous work has addressed the same questions in the contexts we consider (SDG and RRSF).

---

### Author Response · Authors · 2023-11-21
**Eliciting a Response from Reviewers**

Dear Reviewers and ACs,

Thank you for your valuable feedback and suggestions. We hope our rebuttal and the additional contents in the revised manuscript have addressed your concerns. Given the discussion period is drawing to an end, we would be glad to **hear your feedback** and know if our answers clarified all your doubts.

We remain available to respond to any other requests and hope to engage in a **productive discussion**.

Best Regards,

The Authors.

---

### Meta-Review · Area_Chair_1PDx · 2023-12-12

**Metareview:**

This paper proposes enhancing neural image classifiers by performing aggressive data augmentation, intervening on so-called spurious features by using text-to-image generative models. It's worth noting that this whole setup is predicated on already possessing models that do not suffer on account of the spurious associations. I.e., it's not clear that whether this is a method for training a classifiers so much as showing that "your generative model is already, implicitly, the classifier you need". The reviewers gave the paper reasonably thoughtful reviews and although they were late to the conversation, mostly acknowledge / replied to the author's responses. The reviewers generally found the work to be thorough but also to lack any surprising findings. Moreover they point to ample related works that similarly train models using synthetic data augmentation leveraging modern text-to-image models. In short, the paper received 4 reviews with broad consensus that this paper was below the threshold for acceptance. I cannot see any reason to overturn the reviewers judgment.

**Justification For Why Not Higher Score:**

Limited novelty, no champion among the reviewers.

**Justification For Why Not Lower Score:**

N/A

---

### Decision · Program_Chairs · 2024-01-16

Reject